# TNIK signaling imprints CD8+ T cell memory formation early after priming

Carla A. Jaeger-Ruckstuhl [1,2,3,4,7], Magdalena Hinterbrandner[1,2,3,7], Sabine Höpner[1,2], Colin E. Correnti[5], Ursina Lüthi[1,2], Olivier Friedli[3,6], Stefan Freigang [6], Mohamad F. Al Sayed[1,2,3], Elias D. Bührer[1,2,3], Michael A. Amrein [1,2,3], Christian M. Schürch [1,2,6], Ramin Radpour [1,2], Carsten Riether [1,2] & Adrian F. Ochsenbein[1,2✉]

Co-stimulatory signals, cytokines and transcription factors regulate the balance between effector and memory cell differentiation during T cell activation. Here, we analyse the role of the TRAF2-/NCK-interacting kinase (TNIK), a signaling molecule downstream of the tumor necrosis factor superfamily receptors such as CD27, in the regulation of CD8+ T cell fate during acute infection with lymphocytic choriomeningitis virus. Priming of CD8+ T cells induces a TNIK-dependent nuclear translocation of β-catenin with consecutive Wnt pathway activation. TNIK-deficiency during T cell activation results in enhanced differentiation towards effector cells, glycolysis and apoptosis. TNIK signaling enriches for memory precursors by favouring symmetric over asymmetric cell division. This enlarges the pool of memory CD8+ T cells and increases their capacity to expand after re-infection in serial re-transplantation experiments. These findings reveal that TNIK is an important regulator of effector and memory T cell differentiation and induces a population of stem cell-like memory T cells.

[1] Department of Medical Oncology, Inselspital, Bern University Hospital, University of Bern, Bern 3010, Switzerland. [2] Department of BioMedical Research (DBMR), University of Bern, Bern 3008, Switzerland. [3] Graduate School of Cellular and Biomedical Sciences, University of Bern, Bern 3012, Switzerland. [4] Program in Immunology, Fred Hutchinson Cancer Research Center (FHCRC), Seattle, WA 98109, USA. [5] Clinical Research Division, Fred Hutchinson Cancer Research Center (FHCRC), Seattle, WA 98109, USA. [6] Institute of Pathology, University of Bern, Bern 3008, Switzerland. [7]These authors contributed equally: Carla A. Jaeger-Ruckstuhl, Magdalena Hinterbrandner. ✉email: adrian.ochsenbein@insel.ch

During an acute infection, naive CD8$^+$ T cells (T$_N$) are activated, expand, and differentiate into effector cells that mediate host defense and pathogen clearance. These terminally differentiated effector cells (T$_{EFF}$) are usually short-lived. A minor sub-fraction of activated CD8$^+$ T cells develops into memory CD8$^+$ T cells that persist. CD62L(L-selectin)$^{hi}$ C–C chemokine receptor type 7 (CCR7)$^{hi}$ central-memory T cells (T$_{CM}$) are localized primarily in secondary lymphoid organs and have a high proliferative capacity. In contrast, CD62L$^{low}$CCR7$^{low}$ effector-memory T cells (T$_{EM}$) have a high migratory potential, immediate cytotoxic capacity, and provide a first line of defense against reinfection[1].

T-cell differentiation is regulated by several transcription factors. B lymphocyte-induced maturation protein 1 (BLIMP-1), T-box transcription factor (T-BET), Runt-related transcription factor 3 (RUNX3), and NOTCH2 are involved in effector CD8$^+$ T-cell differentiation[2–6]. In contrast, DNA-binding protein inhibitors (ID2/ ID3), forkhead box O1 (FOXO1), and T-cell-specific transcription factor 1 (TCF-1) drive memory formation[7–10]. Importantly, also T-cell receptor (TCR) signaling, cytokines, co-stimulatory signals, and environmental cues determine T-cell expansion and differentiation[4,11–18].

In addition, co-stimulatory receptors including members of the tumor necrosis factor receptor (TNFR) superfamily such as TNFRSF7 (CD27), TNFRSF9 (4-1BB) and TNFRSF4 (OX40) promote the initial expansion and generation of effector T cells, and are implicated in memory formation. TNFR signaling is mediated via TNFR-associated factors (TRAFs). Seven different TRAF proteins have been identified so far[19]. TRAF2 plays a central role in the signal transduction of the majority of TNFRs[20]. Upon activation, TRAF2 is recruited to TNFR intracellular dead domains (DD) or TRAF-interacting motifs (TIM), and regulates NF-kB, MAP kinase, and pro-survival signals[21]. The TRAF2- and NCK-interacting kinase (TNIK) is an important downstream adaptor molecule of TRAF2. TNIK was identified as activator of T-cell factor-4 (TCF-4)[22], a transactivator of the Wnt pathway. TNIK-Wnt signaling is an important oncogenic pathway, and leads to the development of colorectal cancer and the maintenance and expansion of leukemia stem cells[23–26]. However, the role of TNIK in the regulation of CD8$^+$ T-cell biology has not been studied so far.

We therefore developed an inducible TNIK-deficient mouse to study T-cell activation and memory formation during an acute lymphocytic choriomeningitis virus (LCMV) infection. We document that TNIK regulates CD8$^+$ T-cell fate by activating Wnt and other stemness-related pathways and by promoting symmetric over asymmetric cell division. TNIK-deficient T cells preferentially differentiated into short-lived effector cells, while memory T-cell formation was impaired. TNIK-deficient CD8$^+$ T cells lost the capacity to expand in response to antigenic re-stimulation in serial re-transplantation experiments. Moreover, we document that CD27 signaling induces Wnt pathway activation in T cells via the downstream mediator TNIK. These results indicate that TNIK signaling during CD8$^+$ T-cell priming imprints memory formation by regulating T-cell differentiation toward memory T cells.

## Results

**TNIK is dispensable for primary T-cell responses.** We first determined the expression of TNIK in different CD8$^+$ T-cell subsets in LCMV gp33 TCR transgenic mice by ImageStream X. TNIK was expressed at lowest levels in naive p14 T cells (T$_N$; CD62L$^+$CD44$^-$), and its expression increased upon LCMV infection in effector p14 T cells (T$_{EFF}$; CD62L$^-$CD44$^+$). Highest levels of TNIK were found in central memory p14 T cells (T$_{CM}$;

CD62L$^+$CD44$^+$) with slightly lower levels in effector-memory T cells (T$_{EM}$; CD62L$^-$CD44$^+$) (Fig. 1a). Previous studies demonstrated that TNIK acts as Wnt pathway activator and important scaffolding molecule for nuclear trans-localization of active β-catenin[22,26–28]. We found that TNIK is preferentially localized in the cytoplasm of naive p14 T cells, whereas its nuclear expression is increased in effector and memory T cells. Active β-catenin had a very similar expression and distribution pattern as TNIK with a higher nuclear expression in antigen-experienced p14 T cells, especially in T$_{EFF}$ and T$_{EM}$ (Fig. 1a, b).

To evaluate the role of TNIK in CD8$^+$ T cells during an acute viral infection in vivo, we generated a tamoxifen inducible $Tnik^{F/F}$; $UBC$-$Cre^+$ mouse (Supplementary Fig. 1a, b). Knockout efficiency was controlled in PBMCs pre and post treatment on DNA, RNA, and protein level (Supplementary Fig. 1c–e). TNIK deletion ($Tnik^{\Delta/\Delta}$) did not affect steady-state cellularity of T-cell subsets in blood and spleen (SPL) when compared with littermate controls ($Tnik^{WT}$) 10 days after treatment with tamoxifen (Supplementary Fig. 1f, g). Infection of $Tnik^{WT}$ and $Tnik^{\Delta/\Delta}$ mice induced a similar expansion of LCMV gp33-specific CD8$^+$ T cells in the blood and spleen 8 days post infection (p.i.) (Fig. 1c–e). Gp33-specific $Tnik^{WT}$ and $Tnik^{\Delta/\Delta}$ CD8$^+$ T cells differentiated into T$_{EFF}$ cells, expressed comparable levels of Tcf1, granzyme B, and similarly lysed gp33-pulsed target cells in vitro (Fig. 1f–i). We next analyzed early memory and effector fate commitment of gp33-specific $Tnik^{WT}$ and $Tnik^{\Delta/\Delta}$ CD8$^+$ T cells on days 4, 6 (Supplementary Fig. 2a–c) and 8 p.i. (Fig. 1i, j). Frequencies of short-lived effector cells (SLECs; KLRG1$^+$CD127$^-$) and memory precursor effector cells (MPECs; KLRG1$^-$CD127$^+$) in the spleen were comparable (Fig. 1j; Supplementary Fig. 2b). Moreover, T-bet and Eomesodermin (Eomes), two key drivers of cytolytic function[29,30], were similarly expressed on days 4, 6 (Supplementary Fig. 2c) and 8 p.i. (Fig. 1k). These experiments indicate that T-cell activation increases the expression and nuclear translocation of TNIK and β-catenin. However, TNIK deficiency does not impair CD8$^+$ effector T-cell generation.

**TNIK is required for CD8$^+$ T-cell memory formation.** We next evaluated the relevance of TNIK for CD8$^+$ T-cell memory formation after LCMV infection. The frequency of gp33-specific CD8$^+$ T cells in the blood of $Tnik^{WT}$ and $Tnik^{\Delta/\Delta}$ mice was comparable up to 80 days p.i. (Fig. 2a). However, the expression of Tcf1, an essential transcription factor for T cell self-renewal, was lower in $Tnik^{\Delta/\Delta}$ T$_{CM}$ and T$_{EM}$ gp33-specific memory subsets compared to $Tnik^{WT}$ T cells (Fig. 2b). Indeed, the number of gp33-specific CD8$^+$ T cells in the spleen was significantly reduced in $Tnik^{\Delta/\Delta}$ compared to $Tnik^{WT}$ mice (Fig. 2c). In addition, significantly more memory cells were of T$_{EM}$ phenotype with a consequent reduction in the frequency of T$_{CM}$ cells in gp33-specific $Tnik^{\Delta/\Delta}$ mice (Fig. 2d). Fewer $Tnik^{\Delta/\Delta}$ memory CD8$^+$ T cells produced IFNγ, TNFα, and IL-2 after re-stimulation in vitro with PMA/Ionomycin (PMA/I). A similar trend was observed after in vitro re-stimulation with gp33 peptide (Fig. 2e). The mean fluorescence intensity for IFNγ, TNFα, and IL-2 was not different between $Tnik^{WT}$ and $Tnik^{\Delta/\Delta}$ CD8$^+$ T cells, indicating that the production of these cytokines is independent of TNIK expression (Supplementary Fig. 2d).

To assess the re-expansion capacity of LCMV-specific memory T cells, we adoptively transferred (AdTf) FACS-purified gp33-specific $Cd45.2^+$ $Tnik^{WT}$ and $Tnik^{\Delta/\Delta}$ memory T cells into naive $Cd45.1^+$ recipients prior to infection with 200 pfu LCMV (Supplementary Fig. 2e). The re-expansion of AdTf $Tnik^{\Delta/\Delta}$ CD8$^+$ memory T cells 4 days p.i. was significantly reduced (Fig. 2f). Together, these data indicate that TNIK signaling contributes to the generation of functional CD8$^+$ T-cell memory after LCMV infection.

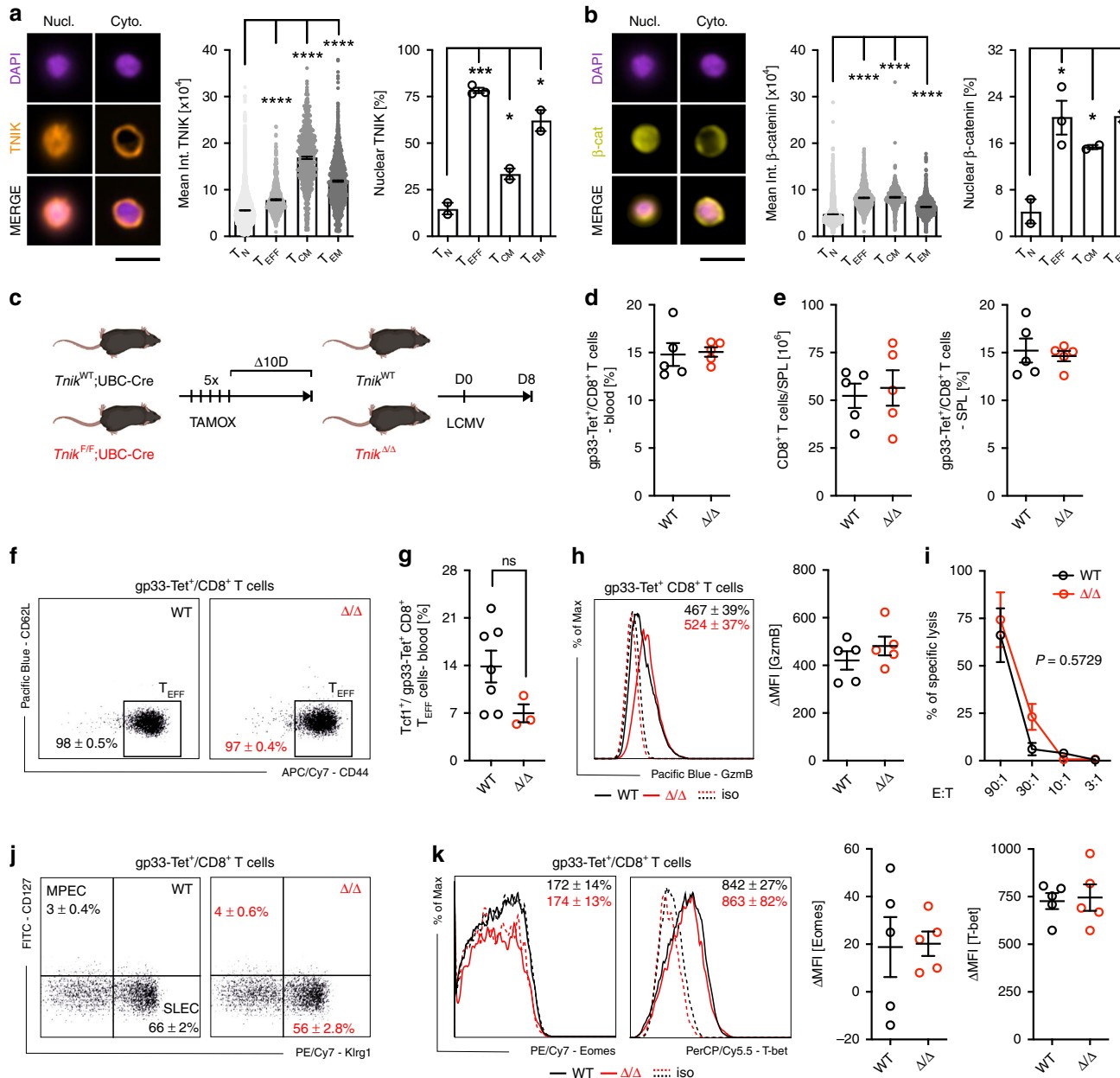

**Fig. 1 TNIK is dispensable for effector T-cell response. a, b** Naive ($T_N$) or activated AdTf p14 T cells isolated from the spleen 6 or 40 days p.i. with $10^4$ pfu LCMV-WE were FACS-sorted, stained for TNIK and β-catenin, and analyzed by ImageStream X (day 6 p.i.: $T_{EFF}$, day 40 p.i.: $T_{CM}$, $T_{EM}$). Single-cell images: nuclear (DAPI) stain, nuclear (Nucl.) vs cytoplasmatic (Cyto.) distribution of TNIK or β-catenin, and overlay (merge). Dotplots: Mean TNIK or β-catenin intensity of $T_N$ ($n = 8638$), $T_{EFF}$ ($n = 1252$), $T_{CM}$ ($n = 630$), $T_{EM}$ ($n = 1308$) and % nuclear localization of $n = 2$–3 sample replicates per cell subset. Scale bar 10 μm. **c** Tamoxifen-induced systemic deletion: $Tnik^{F/F};Ubc-Cre(Tnik^{\Delta/\Delta})$ and control-treated $Tnik^{WT/WT};Ubc-Cre$ ($Tnik^{WT}$) mice were infected i.v. with 200 pfu LCMV-WE. **d–j** Day 8 p.i.: **d** frequency of gp33-Tet$^+$ cells per CD8$^+$ T cells in blood, **e** CD8$^+$ T-cell numbers in the spleen and frequency of gp33-Tet$^+$ cells per CD8$^+$ T cells in the spleen, **f** effector phenotype ($T_{EFF}$, CD44$^+$CD62L$^-$) of gp33-Tet$^+$ CD8$^+$ T cells, **g** frequency of Tcf1 exrepssing $T_{EFF}$ cells, **h** histogram and ΔMFI of granzyme B expression in gp33-Tet$^+$ CD8$^+$ T cell, **i** lysis of gp33 peptide pulsed MC57 target cells by $Tnik^{WT}$ and $Tnik^{\Delta/\Delta}$ CD8$^+$ T cells at different effector/target ratios (E:T), **j** frequency of memory precursor effector cells (MPECs; CD127$^+$KLRG1$^-$) and short-lived effector cells (SLECs; CD127$^-$KLRG1$^+$) within gp33-Tet$^+$ CD8$^+$ T cells, **k** histograms and ΔMFI of intracellular Eomes/T-bet staining in gp33-Tet$^+$ CD8$^+$ T cells. Depicted: $Tnik^{WT}$ (black lines/circles), $Tnik^{\Delta/\Delta}$ (red lines/circles), isotype controls (iso; dashed lines). **d–f**, **h–k** Data are representative for one out of two ($n = 5$) independent experiments. Data are displayed as means ± SEM. Statistics: two-tailed Student's $t$ test, nonsignificant $P > 0.05$, *$P < 0.05$, ***$P < 0.001$, ****$P < 0.0001$. Also see Supplementary Figs. 1 and 2. Source data are provided as a Source Data file.

To distinguish whether TNIK was required for memory imprinting early after activation or during the memory phase and secondary expansion, we conditionally depleted TNIK 20 days after LCMV infection ($Tnik^{\Delta/\Delta20}$, Supplementary Fig. 3a). Only mice with efficient and durable excision of TNIK were included in the experiment (Supplementary Fig. 3b). TNIK depletion after the

initial cytotoxic CD8$^+$ T cell (CTL) priming phase neither changed the frequency of gp33-specific or Tcf1$^+$ $T_{CM}$ and $T_{EM}$ CD8$^+$ T cells in blood nor the number of gp33-specific CD8$^+$ T cells in the spleen (Supplementary Fig. 3c–e). Moreover, the frequencies of gp33-specific $T_{CM}$ and $T_{EM}$ cells were comparable in $Tnik^{WT}$ and $Tnik^{\Delta/\Delta20}$ mice 80 days p.i. in the spleen

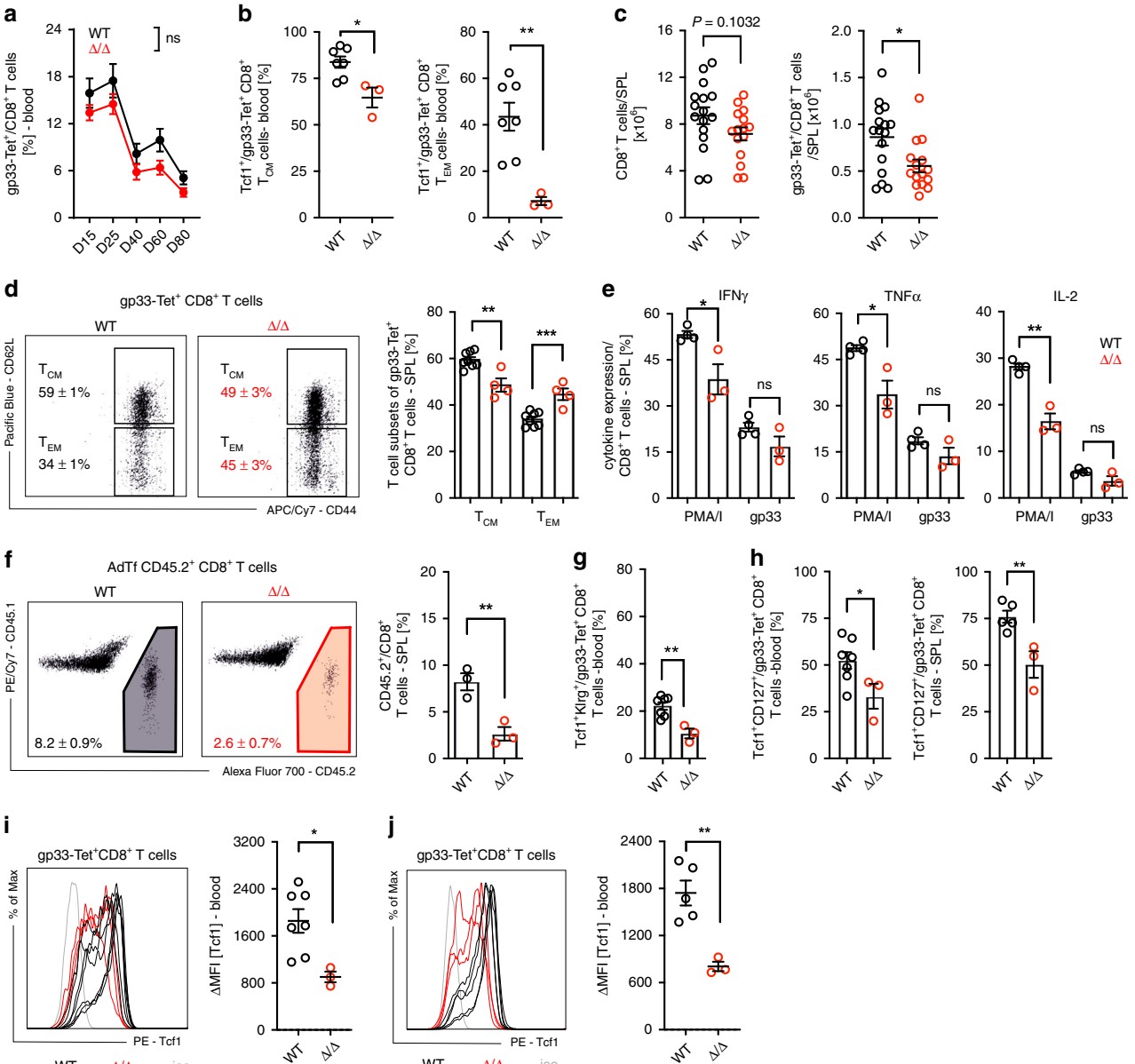

**Fig. 2 Deletion of *Tnik* before priming impairs CD8+ T-cell memory formation. a** Gp33-Tet+ CD8+ T-cell frequency in blood of 200 pfu LCMV-WE-infected *Tnik*^WT and *Tnik*^Δ/Δ mice. **b** Frequency of Tcf1 expressing T_CM (CD44+CD62L+) and T_EM (CD44+CD62L−) gp33-Tet+ CD8+ T cells in blood on day 30 p.i. Mice were killed on day 90 p.i. **c** CD8+ T cells and gp33-Tet+ CD8+ T cells in the spleen. **d** Dotplots and bar graphs showing distribution of gp33-Tet+ CD8+ T_CM and T_EM cell subsets in the spleen. **e** Frequencies of INFγ, TNFα, or IL-2-producing CD8+ T cells after re-stimulation with PMA/Ionomycin or gp33 peptide. **f** AdTf of 3 × 10^4 FACS-purified gp33-Tet+ CD8+ T cells from spleens of day 90 memory mice (*Cd45.2*+) injected into congenic recipient mice (*Cd45.1*+) prior to reinfection with 200 pfu LCMV-WE. Dotplots and bar graphs show relative distribution of endogenous vs AdTf CD8+ T cells 4 days p.i. Frequencies of Tcf1+Klrg+ and Tcf1+CD127+ per gp33-Tet+ CD8+ T-cell subsets in blood or spleen (**g**) 4 days or (**h**) 38 days post rVV-G2 re-challenge. **i, j** Histogram and ΔMFI of Tcf1 expressing gp33-Tet+ CD8+ T cells in blood and spleen day 38 post re-challenge. Depicted: *Tnik*^WT (black lines/circles), *Tnik*^Δ/Δ (red lines/circles), isotype controls (iso; gray lines). **a, d–f** Data are representative for one out of two independent experiments (*n* = 3–4), **c** for three pooled independent experiments (*n* = 15–16), **b, g–j** for one experiment (*n* = 3–7). Data are displayed as means ± SEM. Statistics: **a** two-way ANOVA, **b–j** two-tailed Student's *t* test, nonsignificant *P* > 0.05, *P* < 0.05, **P* < 0.01, ***P* < 0.001. Also see Supplementary Fig. 2. Source data are provided as a Source Data file.

(Supplementary Fig. 3f). Memory CD8+ T cells from *Tnik*^Δ/Δ20 comparably produced IFNγ, TNFα, and IL-2 after in vitro re-stimulation, and expanded similarly in response to LCMV after AdTf into secondary recipients (Supplementary Fig. 3g, h). To evaluate secondary effector and memory fate, mice were re-challenged with recombinant vaccinia virus expressing LCMV-glycoprotein G2 (rVV-G2) 30 days post LCMV infection. Secondary *Tnik*^Δ/Δ but not *Tnik*^Δ/Δ20 effectors showed lower

fraction of Klrg1+Tcf1+ gp33-specific CD8+ T cells compared with *Tnik*^WT 4 days after rVV-G2 infection (Fig. 2g; Supplementary Fig. 3i). Frequencies of CD127+Tcf1+ and maintenance of Tcf1^high gp33-specific CD8+ T cells were reduced in both *Tnik*^Δ/Δ and *Tnik*^Δ/Δ20 mice after antigen re-challenge (Fig. 2h–j; Supplementary Fig. 3j–l). We conclude that TNIK deletion after priming (Δ/Δ20) does not impact primary memory formation and expansion after antigen re-challenge, but affects secondary

memory generation. This indicates that TINIK is required for memory formation during primary and secondary T-cell stimulation and expansion.

**Lack of TNIK in CD8$^+$ T cells impairs memory formation.** TNIK is expressed in various immune cells, including CD8$^+$ T cells, CD4$^+$ T cells, dendritic cells (DCs), B cells, natural killer cells, and innate lymphoid cells (Supplementary Fig. 4a). To analyze the CD8$^+$ T cell intrinsic role of TNIK, we generated p14 TCR transgenic mice harboring a constitutive *Tnik* deletion (Supplementary Fig. 1a). Purified splenic *Cd45.1$^+$ Tnik$^{-/-}$* (KO) and *Cd45.2$^+$* TNIK competent (WT) p14 T cells were adoptively co-transferred (AdCoTf) at a ratio of 1:1 into *Cd45.1$^+$.2$^+$*-recipient mice 1 day prior to infection with 10$^4$ pfu LCMV (Fig. 3a). KO p14 T cells expanded more rapidly reaching >60% of total p14 T cells day 3 p.i. in the spleen. However, the WT to KO p14 T cells ratio was equilibrated on days 7 and 10 p.i. (Fig. 3b). Importantly, more KO p14 T cells underwent apoptosis compared to controls, most significantly during the early contraction phase (Fig. 3c; Supplementary Fig. 4b). In contrast, in vivo Brdu incorporation and CFSE dilution assays did not reveal a significant difference in cell proliferation (Supplementary Fig. 4c, d). Seven days p.i, KO and WT p14 T cells produced similar levels of the effector cytokines IFNγ, TNFα and IL-2 after in vitro re-stimulation with gp33 and expressed comparable amounts of granzyme B and Eomes. The differentiation into MPECs and SLECs was comparable 10 days p.i. (Supplementary Fig. 4e). However, the expression of T-bet was significantly higher in KO vs WT p14 T cells (Fig. 3d–f).

AdTf KO and WT p14 T cells expanded comparably at day 6 p.i. when injected into different recipients. TNIK-deficiency in p14 T cells resulted in an impaired memory formation with reduced frequencies of p14 T cells in blood and spleen (Fig. 3g–i). Nevertheless, the virus was completely eliminated by day 60 p.i. (Supplementary Fig. 4f, g). The majority of KO and WT memory p14 T cells produced IFNγ and TNFα after in vitro re-stimulation with gp33. However, the production of IFNγ per cell was lower in the absence of TNIK (Supplementary Fig. 4h).

To evaluate the re-expansion capacity of memory T cells upon antigen re-exposure, we FACS-purified and CFSE-labeled KO and WT memory p14 T cells from primary recipients for AdTf into secondary recipients. Secondary recipients were subsequently infected with 10$^4$ pfu LCMV. CFSE dilution analysis of AdTf cells in the spleen 3 days p.i. revealed that memory KO p14 T cells re-expand slower than WT p14 T cells, resulting in a significantly lower frequency and number of KO p14 T cells in the spleen (Fig. 3j, k). Sixty days after LCMV infection of secondary recipient mice, KO and WT p14 T cells were FACS-purified from the spleen, and injected at identical numbers into tertiary recipient mice, followed by LCMV infection (Supplementary Fig. 4i). Again, KO p14 T cells expanded less than WT p14 T cells (Fig. 3l).

Thus, while primary peak expansion (day 6 p.i.) of WT and KO p14 T cells was comparable, the capacity to re-expand after a second and third antigen re-challenge dropped significantly (Fig. 3m). Ki67 and AnnexinV staining of the transferred p14 T cells again revealed that the reduced frequency of gp33-specific CD8$^+$ T cells is mainly due to an increase in apoptosis, but not proliferation (Supplementary Fig. 4j, k). These data indicate that TNIK deficiency leads to increased apoptosis of activated T cells during primary infection and impaired formation of functional T-cell memory.

**TNIK regulates differentiation and metabolic reprogramming.** To characterize the molecular pathways involved in TNIK signaling, we performed next-generation RNA-sequencing (NGS)

analysis of WT vs KO naive, effector (D6 p.i.), and memory (D80 p.i.) p14 T cells. Principal component analysis (PCA) revealed that the replicates of naive, effector (D6), and memory (D80) p14 T cells were clustering together. In addition, the analysis indicated that effector and memory populations have a more similar gene expression signature compared to naive T-cell subsets (Supplementary Fig. 5a). In D6 and D80 p14 T cells, a total of 289 and 638 genes were differentially expressed in KO vs WT p14 T cells, respectively (Supplementary Data 1 and 2). Differential gene expression was confirmed by RT-qPCR of selected genes (Supplementary Fig. 5b). Forty-six particular genes were differentially expressed in both effector and memory p14 T cells. These genes were mainly involved in processes regulating metabolism and cell cycle. Interestingly, genes that were higher expressed in KO vs WT p14 T cells D6 p.i. were expressed at lower levels in D80 KO vs WT p14 T cells and vice versa (Supplementary Fig. 5c, d).

Of the differentially expressed genes in D6 p14 T cells, a total of 112 genes were downregulated, and 177 genes upregulated in KO vs WT p14 T cells (Supplementary Fig. 5e). Gene ontology (GO) enrichment analysis revealed that most differentially expressed genes (DEG) were assigned to changes in metabolism, stemness, cell cycle, cell death, immune signaling, and transcription-related processes (Fig. 4a). Gene set enrichment analysis (GSEA) revealed that D6 KO p14 T cells express genes involved in glucose metabolism and oxidative stress response at higher levels than D6 WT p14 T cells, whereas genes involved in fatty acid (FA) metabolism and β-oxidation were expressed at lower levels (Fig. 4b; Supplementary Fig. 5f). From the 80 metabolism-related DEGs in D6 KO vs WT p14 T cells (Supplementary Fig. 5g), 26 were related to catabolic and anabolic processes, mitochondrial biogenesis, oxidative stress, and/or protein synthesis. Key regulatory genes promoting mitochondrial damage and depolarization, oxidative stress, or anabolism (*Noa1, Eif4g1, Polg, Idh2*) were increased in D6 KO p14 cells, whereas genes triggering catabolic processes (*Etfa, Acadsb*) were decreased (Fig. 4c).

Moreover, the driver of terminal effector differentiation *Notch1*[31] as well as its positive regulators *Lfng* and *Aak1*[32,33] were expressed at significantly higher levels in D6 KO vs WT p14 T cells. Similarly, *Ddx6* involved in differentiation[34] and *LIgl2* involved in asymmetric cell division[35,36] were expressed at higher levels in KO p14 T cells (Fig. 4d; Supplementary Fig. 5h). Transcriptional regulators determining T-cell development and function such as *Dnmt1, Hdac7*, and *Pml*[37,38] and cell cycle-related genes were predominantly increased in KO vs WT p14 T cells (Fig. 4d). Moreover, D6 KO p14 T cells showed a pro-apoptotic gene signature (Fig. 4e; Supplementary Fig. 5i). GO analysis revealed that TNIK influences activation of CD8$^+$ T cells and cytokine signaling (Fig. 4a). In accordance, genes set enrichment revealed a higher expression of genes involved in PI3K/Akt and TNF signaling in D6 KO vs WT p14 T cells. Similarly, immune-related genes were mainly expressed at higher levels in D6 KO vs WT p14 T cells (Fig. 4f; Supplementary Fig. 5j, k). In conclusion, NGS analysis revealed that TNIK deficiency in effector CD8$^+$ T cells leads to increased proliferation, differentiation toward effector cells, apoptosis, and a metabolic reprogramming towards increased glycolysis.

To functionally validate the suggested metabolic differences, we performed a Seahorse assay of naive and in vitro-activated WT and KO p14 T cells. We found that naive WT p14 T cells have increased maximal respiratory capacity, spare respiratory capacity, ATP-linked respiration, glycolytic capacity and glycolytic reserve capacity compared with KO p14 T cells (Fig. 4g). After 3 days of in vitro activation using mature bone marrow-derived DCs from H8 transgenic mice (H8-DCs; Supplementary Fig. 5l), which constitutively present gp33-41 on major histocompatibility

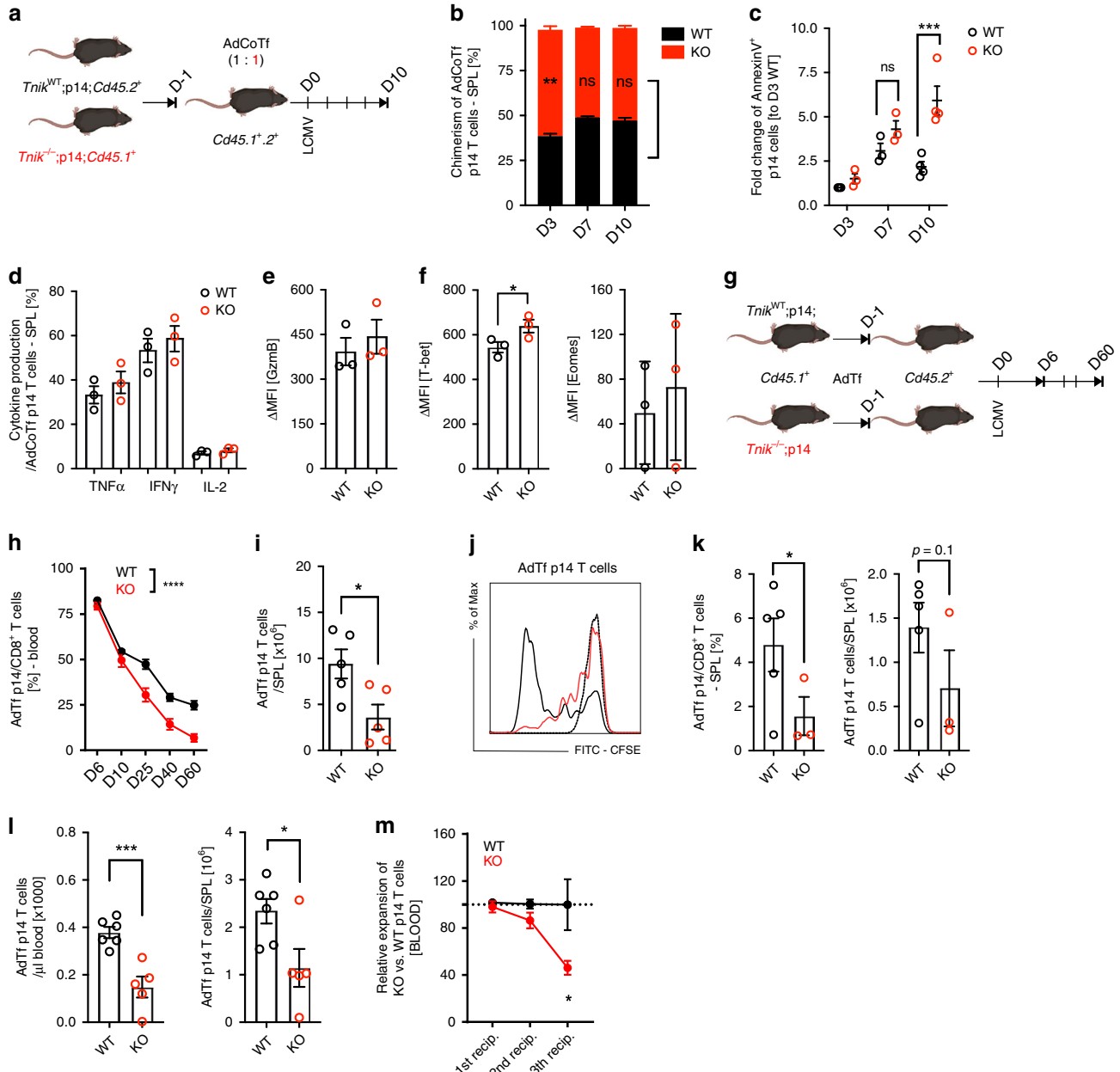

**Fig. 3 Impaired memory formation of TNIK-deficient gp33-specific CD8+ T cells. a** Competitive AdTf (AdCoTf) model: $1 \times 10^5$ splenic $Tnik^{WT}$ (WT; $Cd45.2^+$) and $Tnik^{-/-}$ (KO; $Cd45.1^+$) p14 T cells were AdCoTf into naive $Cd45.1^+Cd45.2^+$ mice previous to infection with $10^4$ pfu LCMV-WE. **b** Chimerism of splenic AdCoTf [(WT;black) vs (KO;red)] p14 T cells ($n = 2$–4 mice/group) 3, 7, or 10 days p.i. **c** Fold change of AnnexinV+ AdCoTf p14 T-cell frequency, normalized to average frequency of day 3 WT ($n = 2$–4 mice per timepoint). **d** Frequency of TNFα+, IFNγ+, and IL-2+ AdCoTf p14 T cells after in vitro re-stimulation with gp33 peptide (7 days p.i.). **e, f** ΔMFI of granzyme B, T-bet, and Eomes in WT/KO effector p14 T cells ($n = 3$ mice per group); normalized to isotype control. **g** Non-competitive AdTf (AdTf) model: $1 \times 10^5$ splenic WT or KO ($Cd45.1^+$) p14 T cells were AdTf into naive $Cd45.2^+$ mice previous to infection with $10^4$ pfu LCMV-WE. **h** P14 T cell frequencies per total CD8+ T cells (blood) at indicated time points. **i** Numbers of AdTf p14 T cells (spleen) day 60 p.i. [representative data from two independent experiments ($n = 5$)]. **j, k** In total, $7 \times 10^5$ CFSE-labeled memory p14 T cells (spleen; pooled from 5 to 7 primary recipients) were injected into congenic secondary recipients, followed by infection with $10^4$ pfu LCMV-WE i.v. **j** CFSE dilution of AdTf p14 T cells 3 days p.i.; non-activated CFSE$^{high}$ (dotted line) p14 T cells (data from one out of two independent experiments). **k** Frequency and absolute numbers of AdTf p14 T cells in the spleen day 3 p.i. **l** Numbers of memory p14 T cells (blood/spleen) 40 days after AdTf of $10^5$ memory p14 T cells into secondary recipients and immunization (200 pfu LCMV; 1st boost). **m** Primary peak expansion rate (1st recipient) of KO p14 T cells normalized to WT T cells ($n = 5$ mice/group) and peak expansion rate of serially re-transferred and re-infected [(2nd recipient ($n = 5$ mice/group); 3rd recipient ($n = 3$ mice/ group)] memory p14 T cells (blood). Depicted: WT p14 (black lines/circles), KO p14 (red lines/circles). Data are displayed as means ± SEM. Statistics: **c, h** two-way ANOVA, **b, d–f, i, k–m** two-tailed Student's t test, nonsignificant $P > 0.05$, *$P < 0.05$, **$P < 0.01$, ***$P < 0.001$, ****$P < 0.0001$. Also see Supplementary Fig. 3. Source data are provided as a Source Data file.

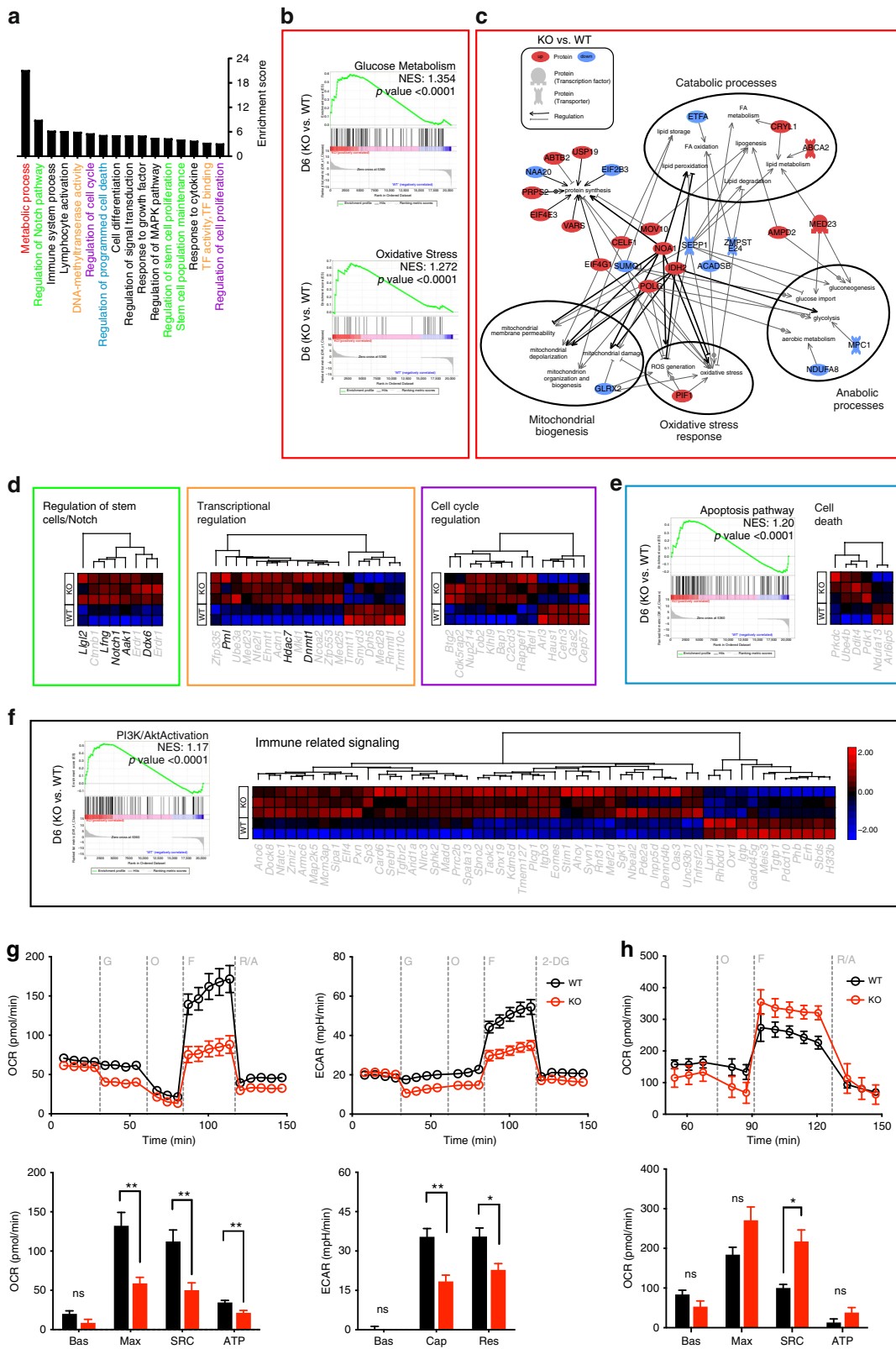

complex class I, KO p14 T cells had a higher spare respiratory capacity and higher glycolytic switch capacity compared to WT p14 T cells (Fig. 4h; Supplementary Fig. 5m). We therefore conclude that TNIK-deficiency compromises intrinsic steady-state metabolic fitness of T cells. The increased glycolytic capacity after activation may be an indication of preferential effector T-cell differentiation, resulting in reduced memory maintenance.

In the memory phase, 303 genes were downregulated and 335 upregulated in KO vs WT p14 T cells (Supplementary Fig. 6a). These genes were associated to similar GO clusters as D6 effector T cells (Fig. 5a). Memory T cells use basal extracellular glucose to support FA β-oxidation and oxidative phosphorylation for survival and homeostasis[39]. GSEA revealed a significant enrichment of basal glucose metabolism and enrichment of FA

**Fig. 4 Gene expression and metabolic flux analysis of *Tnik*^WT and *Tnik*^−/− p14 T_EFF cells. a** GO enrichment analysis of the biological pathways significantly affected in D6 KO vs WT p14 T cells. **b** Gene set enrichment analysis (GSEA) of glucose metabolism and oxidative stress. **c** Gene network and canonical pathway analysis highlighting the regulation and interrelation of metabolic assigned processes. **d** Heatmap of differentially expressed genes assigned to clusters in KO vs WT effector p14 T cells. **e** GSEA of the apoptosis pathway and heatmap of differentially expressed genes in the cell death cluster. **f** GSEA of the PI3K/Akt pathway activation and heatmap of differentially expressed genes in the immune-related signaling cluster. Highlighted genes in the clustered heatmaps are discussed. **g**, **h** Extracellular metabolic flux analysis of (**g**) naive or (**h**) 3 days activated (H8-DCs) WT and KO p14 T cells. Glycolysis (ECAR) and mitochondrial respiration (OCR) were assessed in response to injections of indicated compounds (top), and parameters of glycolysis and OXPHOS were calculated (bottom). Bas, basal glycolysis and respiration; Cap, glycolytic capacity; Res, glycolytic reserve capacity; Max, maximal respiration; SRC, spare respiratory capacity; ATP, ATP-linked respiration; G, glucose; O, oligomycin; F, FCCP; R/A, rotenone/antimycin A; 2-DG, 2-deoxyglucose. Depicted: WT p14 (black circles), KO p14 (red circles). Data are displayed as means ± SEM. Statistics: **g**, **h** two-tailed Student's *t* test, nonsignificant $P > 0.05$, $*P < 0.05$, $**P < 0.01$. Also see Supplementary Fig. 5 and Supplementary Data 1. Source data are provided as a Source Data file.

metabolism and β-oxidation in D80 WT vs KO p14 T cells (Fig. 5b; Supplementary Fig. 6b). Seventy-eight out of the 220 metabolic-related differentially expressed genes were involved in catabolic and anabolic processes, mitochondrial biogenesis, oxidative stress, and protein synthesis (Fig. 5c; Supplementary Fig. 6c). The majority of genes regulating FA β-oxidation and lipid metabolism (*Noa1, Acads, Map3k14, Pck2, Sp2,* and *Acp6*) were downregulated in D80 KO vs WT p14 T cells (Fig. 5c).

In addition, genes involved in T-cell memory formation and maintenance such as the nuclear LEF1 transactivators *Calcoco1* and *Ep300* that are involved in the Wnt pathway and *Bcl11b*, a transcription factor supporting memory differentiation and maintenance[40], were increased in WT vs KO memory p14 T cells (Fig. 5d). Genes involved in proliferation (mitotic centrosome formation: *Cdk5rap2*, G₁ cycle-dependent cell growth: *Cdk6*, actin dynamics: *Rapgef1*) were decreased in KO vs WT D80 memory p14 T cells. In contrast, cell cycle arrest regulators (*Cdk5*) and associated repressors (*Cdk5rap1*) were increased in D80 KO vs WT p14 T cells (Fig. 5d). Similarly, genes associated with immune activation such as *Furin*, a regulator of the transcription factors AP-1, NFAT, and NK-κB, and *Nfkb2, Rora, Stat5a, Lamp1* (CD107α), *Elf4, Cd74* were expressed at higher levels in WT memory p14 memory T cells. In contrast, the transcription factor *Batf* regulating effector fate[41], *Socs4*, impairing T-cell development and homeostasis, and the NFκB inhibitor *Prmt2*, were increased in D80 KO vs WT p14 T cells (Fig. 5e). In conclusion, TNIK-deficient memory T cell display reduced gene signatures in basal metabolic activity, cell cycling, and immune effector function.

**TNIK regulates Wnt signaling and cell division.** Our gene expression analysis and the functional data indicate that TNIK signaling in CD8⁺ T cells reduces terminal differentiation to effector T cells and favors memory commitment early after priming. To study the signaling pathways in more detail, we stimulated p14 T cells with H8-DCs and assessed RNA expression 3–96 h later. TNIK has been shown to activate Wnt signaling in cancer stem cells by enhancing nuclear localization of β-catenin[25,26,42]. Furthermore, Wnt signaling represses Notch signaling[43] and is of importance for T-cell memory formation[44]. We therefore analyzed Wnt target genes early after T-cell activation. *Tcf7, Lef1, Myc, Ctnnb1, Runx1*, and the stemness-related gene *Msi2* were expressed at significantly lower levels in KO vs WT p14 T cells (Fig. 6a; Supplementary Fig. 7a). *Lef1* gene expression was significantly higher in AdTf WT vs KO p14 T cells 48 h p.i., confirming our in vitro data (Supplementary Fig. 7b). However, Wnt target genes were not differentially expressed in the NGS analysis of KO vs WT p14 T cells day 6 p.i., suggesting that Wnt target genes may be induced very early after T-cell stimulation. *Notch1* expression and the expression of genes associated with T-cell effector function (*Ifng, Tbx21, Prdm1*) were expressed at

higher levels in KO p14 T cells 48–96 h after T-cell activation (Fig. 6b; Supplementary Fig. 7c).

In contrast to our in vivo experiments after LCMV infection (Supplementary Fig. 4c, d), KO p14 T cells proliferated slightly more than WT p14 T cells early after activation (Fig. 6c). It is well accepted that a proliferating stem cells can divide symmetrically or asymmetrically to control self-renewal and differentiation[45]. This concept has recently been extended to T cells. Activated T cells undergoing cell division were shown to generate daughter cells with effector and memory fate by asymmetric segregation of cellular determinants[46–48]. We therefore analyzed symmetric division (SD) vs asymmetric division (AD) by Numb segregation 48 h after activation of p14 T cells in vitro and 96 h after LCMV immunization in vivo. TNIK-deficiency significantly enhanced AD in vitro and in vivo (Fig. 6d–f). We therefore hypothesize that the increased frequency of AD versus SD in TNIK-deficient T cells may increase the pool of effector-committed cells and reduce the pool of stem cell/naive-like T cells (T_N, T_EM, T_SCM) (Fig. 6g).

**CD27/TNIK/Wnt signaling favors symmetric cell division.** TNIK interacts with TRAF2, a key mediator of TNFR signaling[20,25,26]. We therefore analyzed whether CD27 signaling activates the Wnt pathway via TNIK and favors SD to enrich for memory T cells. H8-DCs that were used for p14 T-cell activation expressed the CD27 ligand CD70 (Supplementary Fig. 5l). Blocking CD27 signaling by the monoclonal αCD70 antibody (mAb) FR70 in vitro reduced SD to a level comparable with TNIK KO p14 T cells. Importantly, FR70 did not further reduce SD in KO p14 T cells, indicating that CD27 signaling induces SD via TNIK (Fig. 7a). Similarly, in vivo FR70 treatment reduced nuclear β-catenin localization in WT but not in KO p14 T cell day 6 p.i, indicating that Wnt pathway activation is mediated via CD27/TRAF2/TNIK (Fig. 7b). These results suggest that CD27/TNIK signaling induces Wnt pathway activation, SD and favors memory T-cell differentiation. As shown on day 6 p.i., FR70 treatment increased the expression of Eomes and favored differentiation to SLECs in WT, but not KO p14 T cells (Fig. 7c, d).

Asymmetric division gives rise to daughter cells with distinct metabolic characteristics which determines their fate[47]. We next asked whether CD27 signaling during priming of naive human CD8⁺ T cells (T_N) may regulate early metabolic reprogramming via even partitioning of the mitochondria to counteract a qualitative split of daughter cells (effector vs memory differentiation). We therefore activated T_N cells with αCD3/VAR (Varlimumab; αCD27 agonist Ab) or αCD3/αCD28 Ab (TGN1412). αCD3/VAR activation significantly increased symmetric segregation of mitochondria compared to activation via αCD3/αCD28 (Fig. 7e, f; Supplementary Data 3). Importantly, CD3/VAR-activated T_N cells maintained highest TCF-1 expression (Fig. 7g). We next analyzed whether the CD27/TNIK pathway activates Wnt signaling in human CD8⁺ T cells after in vitro re-stimulation. To this end, we FACS-purified

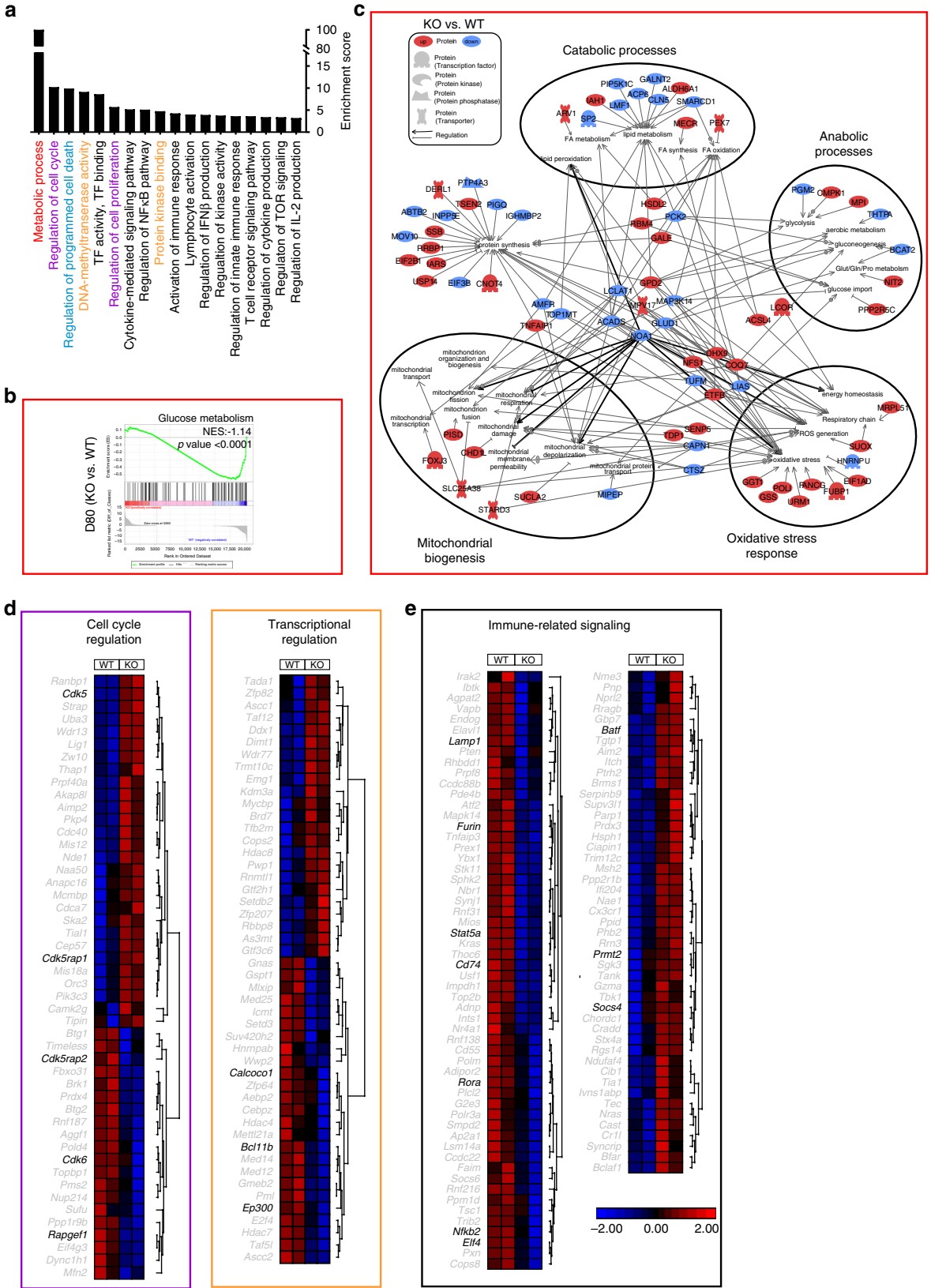

**Fig. 5 Gene expression analysis of *Tnik*^WT and Tnik^−/− memory p14 T cells. a** GO enrichment analysis of the biological pathways significantly affected in D80 KO vs WT p14 T cells. **b** GSEA for glucose metabolism. **c** Gene network and canonical pathway analysis highlighting the regulation and interrelation of metabolic assigned processes. **d**, **e** Heatmap of differentially expressed genes assigned to specific clusters in KO vs WT memory p14 T cells. Highlighted genes in the clustered heatmaps are discussed. NES Normalized Enrichment Score. Also see Supplementary Fig. 5 and Supplementary Data 2. Source data are provided as a Source Data file.

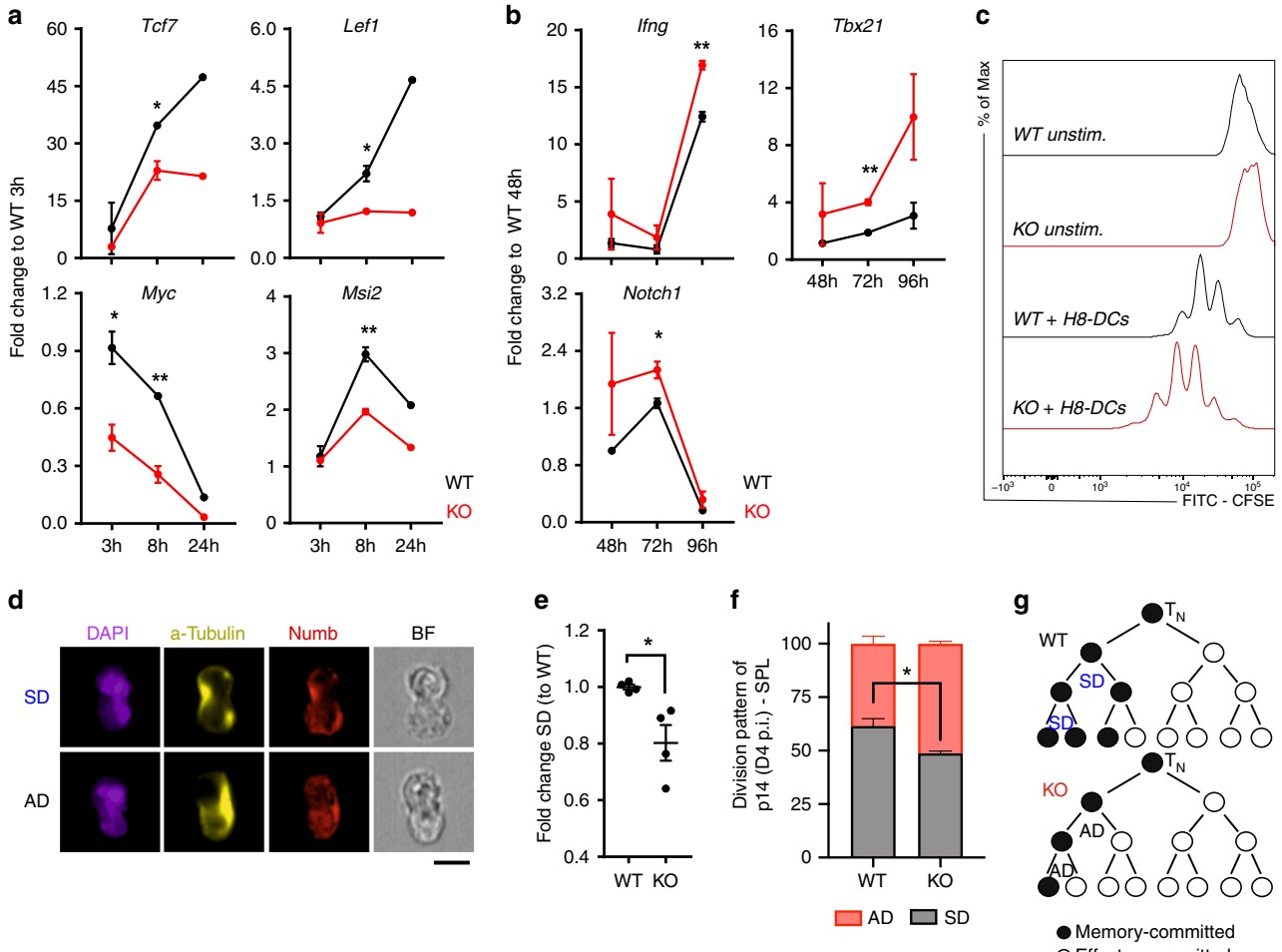

**Fig. 6 TNIK signaling activates the Wnt pathway and favors symmetric cell division. a, b** WT (black) and KO (red) p14 T cells were co-cultured with mature bone marrow-derived H8-DCs (H8-DCs) (1:1 ratio). Relative gene expression of indicated genes in FACS-purified p14 T cells is shown 3–96 h post activation. One out of two independent experiments is shown. **c** CFSE dilution of WT or KO p14 T cells after 72 h of co-culture with mature H8-DCs. Representative data from two independent experiments is shown. **d** ImageStream X analysis of Numb segregation in mitotic p14 T cells after 48 h of co-culture with H8-DCs; scale bar 10 μm. Analysis includes a total of 375–499 dividing mitotic cells. Representative nuclear stain (DAPI), α-tubulin stain, Numb stain and bright field (BF) depict symmetric division (SD) vs asymmetric division (AD); scale bar 10 μm. **e** Numb segregation per each culture replicate (n = 2–3) and frequencies of SD vs AD were assessed. Fold change of SD frequency normalized to average of WT control group is depicted. Pooled data from two independent experiments is shown (two replicates per timepoint). **f** ImageStream X analysis of Numb segregation in mitotic AdTf p14 T cells isolated of spleens 96 h p.i. Frequency of AD vs SD of total dividing WT and KO p14 T cells is shown. Analysis includes a total of 192 (KO) and 186 (WT) dividing mitotic cells from three pooled mice per replicate (n = 4). **g** Model of cell fate in KO vs WT T cells. Three cell division steps are observed within 48 h post activation. Data are displayed as means ± SEM. Depicted: WT p14 (black lines/circles), KO p14 (red lines/circles). Statistics: **a, b, e, f** two-tailed Student's t test, nonsignificant P > 0.05, *P < 0.05, **P < 0.01. Also see Supplementary Fig. 6. Source data are provided as a Source Data file.

CD27+ and CD27− subpopulations from an in vitro expanded HIV-1 gag-specific CD8+ T clone[49] (Fig. 7h). Purified CD27+ and CD27− T cells were activated with αCD3 mAb in the presence of blocking αCD27 mAb or IgG control. Absence of CD27 on T cells or the addition of αCD27 mAb significantly reduced nuclear localization of TNIK and β-catenin (Fig. 7i). A TCF-1/LEF7 reporter assay confirmed a significantly higher Wnt activity in CD27+ T cells compared with CD27− T cells (Fig. 7j). Taken together, CD27 co-stimulation supports early metabolic fitness via the TNIK/Wnt signaling pathway in murine and human CD8+ T cells.

## Discussion

An acute infection leads to an increased demand of different immune cells. Myeloid cells are produced in the bone marrow[50] by hematopoietic stem cells (HSCs) and the process of differentiation versus self-renewal during emergency hematopoiesis is tightly regulated[51]. In contrast, CD8+ T cells serve the increased demands during a viral infection by clonal expansion[52]. Similarly to HSCs, the balance between differentiation to full effector cells and the maintenance of cells with the capacity to persist long-term is crucial to eradicate the pathogen and to allow memory formation. Indeed, it became increasingly recognized that conserved pathways regulate HSC and T-cell maintenance, introducing a new identity of memory T cells with stem-cell-like features[53].

TNIK regulates stemness in small intestine crypts and supports colorectal cancer formation[54]. Similarly, TNIK signaling induces proliferation and self-renewal of leukemia stem cells[25,26]. Here, we document that TNIK favors the generation of memory T cells during CD8+ T-cell activation while reducing differentiation to effector cells. TNIK-deficient effector T cells were characterized by an increased apoptotic rate and glycolysis. In contrast, TNIK-deficient memory T cells expanded less efficiently upon

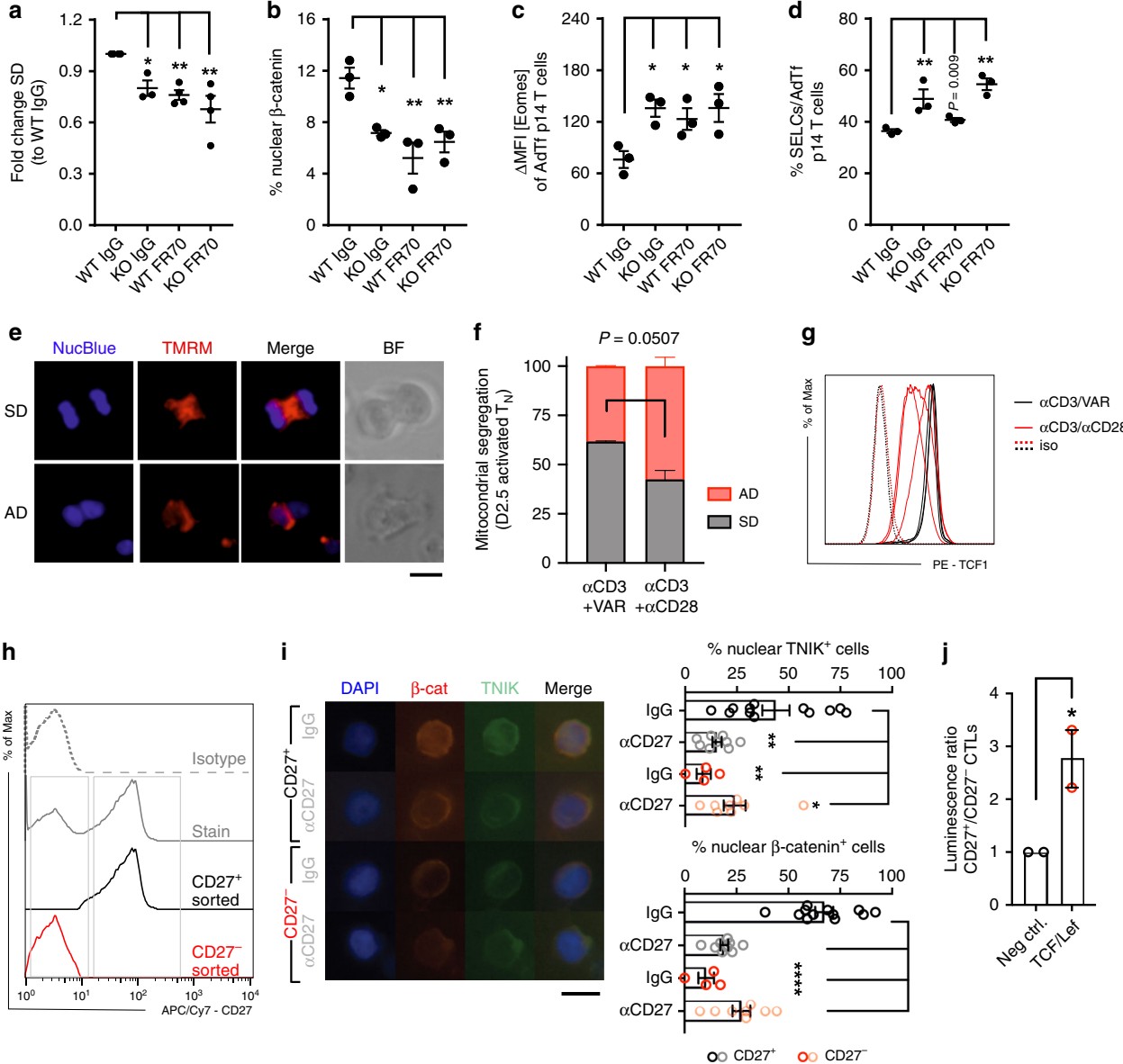

**Fig. 7 TNIK is a CD27 downstream adaptor molecule and regulates Wnt signaling. a** ImageStream X analysis of Numb segregation in mitotic WT and KO p14 T cells after 48 h of co-culture with H8-DCs in the presence of 20 μg ml⁻¹ blocking FR70 Ab or IgG (175–333 mitotic cells). Numb segregation per culture replicate was analyzed, and frequencies of asymmetric division (AD) vs symmetric division (SD) were assessed. Fold change of SD normalized to WT IgG control is depicted. Pooled data from two independent experiments (1–2 replicates per timepoint). **b–d** In total, $10^5$ $Cd45.1^+$ $Tnik^{WT}$ (WT) or $Tnik^{-/-}$ (KO) p14 T cells from the spleen were AdTf into naive $Cd45.2^+$ mice previous to infection with $10^4$ pfu LCMV-WE. Mice were injected i.p. with 300 μg FR70 mAb or IgG on days −1/+2/+5. **b–d** Analysis day 6 p.i. ($n = 3$ mice per group). **b** FACS-sorted p14 T cells stained for β-catenin. Dotplot shows frequency of nuclear localization analyzed by ImageStream X. **c** ΔMFI of Eomes and **d** frequency of $KLRG1^+CD127^-$ p14 SLECs (spleen). **e, f** Live imaging of dividing αCD3/VAR ($n = 183$) or αCD3αCD28 ($n = 170$) in vitro-activated human naive $CD8^+$ T cells ($T_N$) (two donors, two independent experiments): nucleus (NucBlue) and mitochondira (TMRM) stain, merge and brigh field (BF) of representative mitotic cells; scale bar 10 μm. Frequency of AD vs SD of total ($n = 170-182$) dividing cells is shown. **g** Histogram showing TCF-1 expression of αCD3/VAR (black lines) or αCD3αCD28 (red lines) activated $T_N$ cells and isotype controls (iso; dashed lines). **h** Histogram showing CD27 expression (gray solid line), isotype control (gray dotted line), purified $CD27^+$ (black line), and $CD27^-$ (red line) HIV gag-specific human $CD8^+$ T cell clone. **i** $CD27^+$ (black/gray dots) and $CD27^-$ (red/light red dots) T cells cultured under 0.3 μg ml⁻¹ OKT3 and 10 μg ml⁻¹ CD27 blocking mAb or IgG. Percentage of cells positive for nuclear TNIK or β-catenin is depicted (three independent experiments; 4–12 replicates; 60–79 cells per replicate); scale bar 10 μm. **j** TCF/LEF lentiviral reporter assay: luminescence ratio ($CD27^+/CD27^-$) relative to negative control lentiviral particles. **h, i** Representative results of two independent experiments depicted. Data displayed as means ± SEM. Statistics: **a–d, i** one-way ANOVA, **f, j** two-tailed Student's t test, nonsignificant $P > 0.05$, *$P < 0.05$, **$P < 0.01$, ****$P < 0.0001$. Source data are provided as a Source Data file.

reinfection, and lost the capacity to expand and engraft in serial re-transplantation experiments, a classical assay used to assess stem-cell function of HSCs[55].

Upon activation, TNIK is recruited to the nucleus to complex with TCF-4/β-catenin and induce the canonical Wnt signaling pathway[22,56]. We documented a reduced expression of Wnt target genes in TNIK KO p14 T cells early after activation in vitro and 48 h after infection in vivo. In contrast, Wnt pathway activation was not detected in the NGS analysis at the peak of the T-cell expansion in vivo. This may indicate that TNIK-Wnt activation is a very early event after T-cell activation. Alternatively, high abundance of effector cells day 6 p.i. may prevent the detection of differently regulated genes in the low abundant memory T-cell precursors by using bulk RNAseq. However, Notch1 and other molecules associated with differentiation to effector cells such as Lfng and Aak1 are upregulated in TNIK KO effector p14 T cells. Notch and Wnt pathways are highly conserved interrelated signaling pathways that reciprocally control cell fate[57]. In CD8[+] T cells, Notch signaling promotes effector differentiation while inhibiting the signaling pathways promoting memory T-cell formation[6]. Moreover, Notch activates the PI3K/Akt/mTOR pathway that is critical for metabolic conversion to glycolysis, allowing rapid proliferation and acquisition of effector function by T cells[47]. Importantly, GSE analysis of TNIK-deficient effector cells revealed a significantly higher expression of genes involved in the PI3K/Akt pathway, suggesting that Akt and mTOR kinases contribute to the increased glycolysis.

Wnt signaling favors the differentiation into memory precursor cells[10]. The Wnt target genes Tcf7 and Lef1 are preferentially expressed in $T_N$ and in $T_{CM}$, but not in $T_{EFF}$ cells[58]. Moreover, activation of the Wnt pathway in vitro suppressed the antigen-induced expression Eomes and inhibited differentiation to effector T cells. This arrested differentiation favored the generation of $T_{CM}$ and T memory stem cells that are characterized by a high proliferative capacity upon TCR re-stimulation[53,59]. Further, Tcf7-deficient mice lack CD8[+] memory precursor T cells, and Tcf7-deficient T cells have an impaired capacity to expand upon reinfection[44]. Interestingly, the phenotype of TNIK-deficient CD8[+] T cells is very similar to these earlier results analyzing the Wnt pathway in T-cell memory formation. TNIK signaling reduced the expression of Eomes and the differentiation to SLECs in vitro, and KO p14 T cells had a severe impairment in secondary expansion.

We recently showed that TNIK signaling favors symmetric (SD) over asymmetric (AD) cell division in leukemia stem cells[25]. Whether memory T cells emerge in a linear path from effector cells or whether memory precursors are induced early after activation is still a matter of debate[60–62]. Several studies documented an asymmetric segregation of cellular components such as IFNγ receptor, mammalian target of rapamycin complex 1 (mTORC1) kinase, and the transcription factor c-Myc in the first cell division steps, supporting an early divergence of effector and memory T-cell precursors[47,48,63]. We now document that absence of TNIK increases the frequency of T cells undergoing AD in the first three division steps after activation. This promotes differentiation to effector cells and a higher glycolytic activity. Terminally differentiated effector cells usually kill target cells an then undergo apoptosis. Indeed, we documented an increased apoptosis rate of CD8[+] T cells after expansion in the absence of TNIK.

Since TNIK is a downstream adaptor molecule of TRAF2[64], we analyzed whether the TRAF2 signaling TNFR CD27 induces TNIK/Wnt signaling. The importance of the CD27/TNIK/Wnt signaling pathway has been documented before in leukemia stem cells[25,26]. Similarly, CD27 co-stimulation induced nuclear TNIK translocation and Wnt pathway activation in murine and in human CD8[+] T cells. This suggests that the CD27/TNIK/Wnt

pathway favors memory formation early after CD8[+] T-cell activation. In addition, CD27 signaling favored symmetric cell division and improved the metabolic capacity of the daughter cells by regulating the partitioning of the mitochondria. However, it is well documented that CD27 signaling in memory CD8[+] T cells is involved in the maintenance of T-cell memory and potentiates autocrine IL-2 production and re-expansion capacity[49,65,66]. Importantly, TNIK depletion day 20 p.i. did not alter the re-expansion capacity of CD8[+] T cells but secondary memory maintenace. This suggests, that CD27 ligation on memory T cells supports re-expansion independently of TNIK, most likely by activating NFκB and c-Jun N-terminal kinases[67,68].

In summary, this study identifies TNIK as an important regulator of effector and memory T-cell differentiation early after T-cell priming and provides a link between TNFR signaling and the Wnt pathway in T cells.

## Methods

**Mice.** C57BL/6J mice (BL/6, Cd.45.2[+]) were obtained from Charles River Laboratories. Cd45.1[+] congenic BL/6 mice, p14 TCR transgenic, and H8 gp33-transgenic mice were provided from the Institute for Laboratory Animals (Zürich, Switzerland). Inducible and constitutive TNIK knockout mice were generated and maintained locally. A TNIK knockout first allele (KOMP Repository; Project ID: CSD39738) was inserted into embryonic stem (ES) cells by homologous recombination. ES cells were injected into blastocystes and implanted following to foster mother. Founder mice were bred and backcrossed to a C57BL/6J background. Founder mice were further crossed with Flp[+/+] mice, where the non-expressing allele was converted into a conditional allele upon recombination of the FRT sites. Mice harboring the floxed Tnik allele Tnik[F/F] were used for Cre-recombination-based generation of inducible or constitutive knockout mice. By crossing Tnik[F/F] with inducible promotor driven B6.Cg-Tg(UBC-cre/ESR1)1Ejb/J mice (Ubc-Cre[ERT2]; Jackson Laboratory), Tnik[F/F];UBC-Cre or littermate controls Tnik[WT/WT];Ubc-Cre were generated. Genotyping primers (Supplementary Table 1) were designed by KOMP Repository (Design ID: 49289). Per oral (p.o.) administration of tamoxifen (200 mg kg[−1] day[−1]) on 5 consecutive days allowed Cre-mediated TNIK deletion. By crossing Tnik[F/F] with C57BL/6-Tg(Zp3-cre)93Knw/J mice (Jackson Laboratory), constitutive Tnik[−/−] or littermate controls Tnik[+/+] were generated. P14 TCR mice were crossed with Tnik[−/−] mice. Based on SNP analysis, in house generated TNIK mice were 97.65–99.97% C57BL/6 background. All mice used in this study were maintained under specific pathogen-free conditions and entered the experiment at the earliest age of 6 weeks. Animal experiments received ethical approval by the local experimental animal committee of the Canton of Bern, and were performed according to Swiss laws for animal protection. All mouse illustrations are creatd with Biorender.com.

**Cell lines.** The human lymphoblastoid B-cell line TM-LCL was generated in the Lab of Dr. Phil Greenberg at the Fred Hutchinson Cancer Research Center, Seattle, USA. The murine fibrosarcoma cell line MC57 was purchased from ATCC.

**Human samples.** Human HIV-1 Gag-specific CD8[+] T-cell clones were generated at the Fred Hutchinson Cancer Research Center, Seattle, USA, via in vitro expansion of peripheral blood mononuclear cells (PBMCs) with UV-inactivated vac/gag-infected PBMC-derived stimulator cells[49,69]. Human naive CD8[+] T cells were derived from healthy donor PBMCs and isolated following the instructions of EasySep Human Naive CD8[+] T Cell Isolation Kit (STEM CELL). All uses of human material have been approved and reviewed by the Institutional Review Office at the Fred Hutchinson Cancer Research Center. All recruited volunteers provided written informed consent to participate as donors of peripheral blood mononuclear cells.

**Viral infection.** LCMV strain WE was provided by R.M. Zinkernagel (University of Zürich, Zürich, Switzerland) and propagated on L929 fibroblasts[70,71]. Tnik[F/F]; Ubc-Cre mice and littermate controls were infected with 200 plaque-forming units (pfu) LCMV-WE. Alternatively, $1 \times 10^5$ MACS-purified p14 CD8[+] T cells from p14;Tnik[−/−] mice or littermate controls were adoptively transferred to congenic BL/6 recipients. Eighteen hours later, mice were infected with $10^4$ pfu LCMV-WE. Re-challenge experiments were performed using $2 \times 10^6$ pfu recombinant vaccinia virus expressing LCMV-glycoprotein G2 (rVV-G2)[72]. Alternatively, FACS-purifed gp33-specific CD8[+] or p14 T cells were AdTf into secondary recipient mice followed by infection with 200 or $10^4$ pfu LCMV-WE, respectively.

**Plaques-forming assay.** LCMV virus titers were determined by plaque-forming assay on adherent MC57 fibroblast cell lines. After absorbance of virus from liver homogenates, adherent cells were fixed with PBS 4% PFA, permeabilized with 0.5% Triton X-100 balanced salt solution, and stained with anti-LCMV antibody[73].

**Serial re-transplantation and reinfection**. Memory mice were harvested ≥ 40 days p.i., and splenic CD8[+] T cells were MACS-purified using LS-MACS columns. Gp33 epitope-specific polyclonal CD8[+] T cells or monoclonal p14 CD8[+] T cells were FACS-sorted using tetramers or anti-TCR Vα2 antibody, respectively. For first reinfection, $3 \times 10^4$ tetramer-specific CD8[+] memory T cells or $1 \times 10^5$ p14 memory T cells were adoptively transferred into secondary congenic recipient mice. For secondary infection, $3 \times 10^4$ p14 cells were re-transferred. In first and second re-infection, mice were injected with $10^4$ pfu LCMV.

**Antibodies and flow cytometry**. Quantitative flow cytometric analysis was performed using standard procedures. The following anti-mouse antigen-specific fluorochrome-conjugated antibodies were used: CD8 (53-6.7), CD19 (6D5), CD4 (GK1.5), CD11c (N418), CD80 (16-10A1), CD86 (GL1), CD45.2 (104), CD45.1 (A20), CD62L (MEL-14), CD44 (IM7), KLRG1 (2F1/KLRG1), CD127 (A7R34), CD70 (FR70), TCR Vα2 (B20.1), CD107α (1D4B) (BioLegend). For intracellular staining TNFα (MP6-XT22), IFNγ (XMG1.2), IL-2 (JES6-5H4), Eomes (Dan11-mag) (eBioscience), T-bet (4B10), GranzymeB (2E7) (BioLegend), Tcf7/Tcf1 (533-966) (BD Pharmingen)-directed antibodies, and the corresponding isotype controls were used. The following anti-human antigen-specific fluorochrome-conjugated antibodies were used: CD27 (MT-271), CD8 (HIT8a), and Tcf7/Tcf1 (533-966). Fluorescent-labeled tetramers with H-2D[b] LCMV gp33-41 (KAVYNFATC) were purchased from MBL International Corporation. Data acquisition was performed using a LSR Fortessa or a LSR II SORP (BD Biosciences) flow cytometer. Cell-sorting was performed using FACS Aria or FACS Aria III (BD Bioscience). Analysis was done using FlowJo (V.10.0.8, TreeStar, Inc.) software. For in vitro stimulation of human CD8[+] T cells, αCD3 antibody (OKT3, BioXCell), anti-human CD27 antibody (Varlimumab, 1F5, produced at FHCRC), and anti-human CD28 antibody (TGN1412, produced at FHCRC) were used. For in vitro blocking, anti-human CD27 antibody (1A4CD27, Beckman Coulter) was used. For flow imaging and immunofluorescence, primary rabbit-α-TNIK (Santa Cruz Biotechnology), rabbit-α-Numb (Abcam), mouse-α-active-β-catenin (Merck Millipore), mouse-α-alpha tubulin (Abcam) and secondary goat-α-rabbit Alexa Fluor 546, goat-α-rabbit Alexa Fluor 647 (Abcam), goat-α-mouse Alex Fluor 546 (Invitorgen), and biotinylated goat-α-rabbit (Invitrogen) were used. For in vitro blocking of CD27 signaling, CD70 (FR70; BioXCell) or control rat IgG (SIGMA) was used.

**Chemicals, peptides, and recombinant proteins**. See Table 1 for a list of used reagents.

**ImageStream X analysis**. FACS-sorted cells were fixed in 4% paraformaldehyde (PFA) kept for 10 min at 4 °C. Alternatively, 72 h anti-CD70 (FR70, 20 µg ml[−1]) or IgG-treated in vitro bone marrow-derived H8-DC activated p14 cells were fixed in freshly prepared Fix/Perm reagent provided by the FoxP3 staining kit (Thermo Scientific) for 1 h at 37 °C. Then, cells were washed 2 × 5 min with 1× Wash Buffer (Dako) or 1× Perm/Wash buffer (Thermo Scientific). Fc receptor blocking was performed (Innovex biosciences) for 30 min at RT. After another washing step, cells were incubated with the primary anti-TNIK (D-16, Santa Cruz Biotechnologies), anti-β-catenin (Merck Millipore), anti-α-tubulin (Abcam), or anti-Numb (Abcam) antibodies in diluent (Dako) or in 1× Perm/Wash buffer (Thermo Scientific) overnight at 4 °C. The next day, another washing step was performed in 1× Wash Buffer (Dako) or 1× Perm/Wash buffer (Thermo Scientific) prior to incubation with the secondary antibody for 1 h at RT in the dark. Cells were then washed again twice in PBS. Lastly, DAPI (Roche) nuclear staining was performed, incubating the cells for 5 min at RT in the dark. After the last washing step in PBS, cells were acquired at the ImageStream X Mark II imaging flow cytometer (Amnis) and analyzed using INSPIRE and IDEAS software. Dividing CD8[+] T cells were extracted using a Mitosis Analysis Wizzard provided by IDEAS software. Selected dividing cells were double proved by two independent researchers according to nuclear DAPI stain. Using an intensity mask for Numb, symmetric (<1.8-fold) versus asymmetric (>1.8-fold) segregation was determined in dividing cells. Dividing cells that failed intensity calculation were assigned manually.

**Immunofluorescence staining of human T-cell clones**. FACS-sorted cells were put on glass slides, fixed in 4% PFA, and treated with blocking solution. Staining was performed for 2 h at RT using primary anti-TNIK (D-16, Santa Cruz Biotechnologies) and anti-β-catenin (Merck Milllipore) antibodies. After washing, cells were stained with the corresponding biotinylated secondary antibodies (Invitrogen). Following a Streptavidin labeling step for 1 h at RT, detection of biotinylated secondary antibody was enabled. Lastly, DAPI (Roche) nuclear staining was performed. Nuclear/cytoplasmic localization of TNIK and β-catenin was visualized by an Eclipse E800 fluorescence microscope and Nikon DS-Ri1 camera NIS Elements BR 3.0/3.2 software.

**Live imaging of mitochondrial separation**. In total, $2 \times 10^6$ CD8[+] $T_N$ cells were activated with plate coated αCD3 (5 µg ml[−1]) and αCD27 (VAR; 5 µg ml[−1]) or αCD28 (2 µg ml[−1]) agonistic antibody in vitro. After 2.5 days of activation, cells were washed in pre-warmed CTL media [RPMI, 10% heat-inactivated human AB sera, 2% L-glutamin (4 mM), 1% penicillin/streptomycin, 0.01% β-mercaptoethanol (0.5 M)] supplemented with 50 U ml[−1] human IL-2. Cells were then incubated with Image-iT[TM] TMRM Reagent (Thermo Fisher) and NucBlue[TM] Live Ready-Probes[TM] Reagent (Thermo Fisher) for 30 min at 37 °C under standard culture conditions. Subsequently, images were aquired with EVOS M5000 Imaging System using a 40X objective. TMRM apportioning in deviding daughter cells was visually assessed.

**Human CD8[+] T-cell clone expansion**. Human HIV gag-specific CD8[+] T-cell clones were cultured as previously described[49]. In vitro re-expansion was performed in CTL media supplemented with 50 U ml[−1] recombinant human IL-2, 0.03 µg ml[−1] OKT3 and irradiated lymphoblastoid feeder cells (LCL). Half-media change was performed on days 1, 5, 7, and 10 after activation. FACS-sorted CD27[+] and CD27[−] CD8[+] T cells were further used for experiments and cultured in the presence of 0.3 µg ml[−1] αCD3 (OKT3) and 10 µg ml[−1] CD27 blocking mAb or isotype control.

**Proliferation assay and cell cycle analysis**. *In vivo proliferation assay*: FACS-sorted p14 memory T cells were labeled with 5 µM carboxyfluorescein succinimidyl ester (CFSE) for 10 min at 37 °C, and washed twice with PBS (5% FCS). In total, $7 \times 10^5$ labeled p14 cells were transferred into congenic recipient mice that were infected with $10^4$ pfu LCMV-WE 18 h later. Frequency of transferred T cells was analyzed 3 days p.i. Alternatively, BrdU (2 mg 200[−1] µl[−1]) was injected (i.p.) into LCMV-immunized mice to determine 3 h incorporation rate and cell cycle state of expanding p14 T cells using the APC BrdU Flow Kit (BD Biosciences) and 7AAD labeling.

*In vitro proliferation assay*: In all, $10^5$ CFSE-labeled naive WT/KO p14 cells were co-cultured with bone marrow-derived H8-DCs that were generated as previously described[74].

*Ex vivo proliferation*: Intracellular staining was performed using the Ki67 staining kit (BD Bioscience).

**Cytokine release assay**. *In vitro*: Splenocytes were incubated for 4–5 h at 37 °C with Phorbol-12-myristate-13-acetate (PMA, 100 ng ml[−1]) and Ionomycin (1000 ng ml[−1]) or MHC-I-specific LCMV-glycoprotein peptides gp33 (aa 33–41, KAVYNFATC) was purchased from NeoMPS SA. Protein accumulation in the

### Table 1 Chemicals, peptides, and recombinant proteins.

| Product | Provider |
| --- | --- |
| LCMV-gp$_{33-41}$ (KAVYNFATC) tetramer H-2 D[b] | MBL, Woburn, MA, USA |
| LCMV-gp$_{33-41}$ (KAVYNFATC) peptide | NeoMPS SA, Strasbourg, France |
| rhIL-2 | Prospec, Israel |
| rmIL-2 | Prospec, Israel |
| Phorbol-12-myristate-13-acetate (PMA) | Sigma Aldrich, St. Louis, MO, USA |
| Ionomycin | Santa Cruz Biotec., Dallas, TX, USA |
| Brefeldin A | Sigma Aldrich, St. Louis, MO, USA |
| DAPI | Roche, Basel, Switzerland |
| Tamoxifen | Sigma Aldrich, St. Louis, MO, USA |
| Phosphatase inhibitor | Thermo Fisher, Waltham, MA, USA |
| Streptavidin | BD Biosciences, San Jose, CA, USA |
| Fc receptor blocking reagent | Innovex Biosciences, Richmont, CA, USA |
| Image-iT[TM] TMRM reagent | Invitrogen, Life Technologies, CA, USA |
| NucBlue[TM] Live ReadyProbes[TM] Reagent | Thermo Fisher, Waltham, MA, USA |

endoplasmatic reticulum was induced by Brefeldin A (5 µg ml$^{-1}$, Sigma Aldrich). Culture media was supplemented with rmIL-2 (12.5 U ml$^{-1}$).

*Ex vivo*: Brefeldin A (1.25 mg) was injected i.v. 6 h before analysis. Staining for cell surface antigens was performed for 30 min at 4 °C. After an additional washing step, cells were permeabilized using BD Cytofix/Cytoperm solution (BD Bioscience) for 30 min at RT in the dark. Intracellular staining was performed in BD Perm/Wash buffer (BD Bioscience) for 30 min at 4 °C.

**Cytotoxicity assay**. In total, $1 \times 10^4$ gp33-peptide pulsed MC57 target cells were co-incubated with CD8$^+$ effector T cells from day 8 LCMV-infected mice at indicated ratios (1:3, 1:10, 1:30, and 1:90) in a 96-V-bottom plate. Cultures were incubated for 4–5 h at 37 °C. In all, 96-well plates were centrifuged, and supernatants were transferred into non-transparent 96-well plates. Resazurin reaction mixture was added (1:1) to cell supernatants. The plate was incubated for 10 min at 37 °C protected from light. Fluorescence reaction was recorded using a microplate reader (TecanReader). Killing efficiency of effector T cells was calculated and normalized to non-lysate-treated conditions. Maximal killing efficiency (100%) was set by lysate control measurements.

**Extracellular metabolic flux analysis**. The metabolic activity (glycolysis and mitochondrial respiration) of p14 T cells was determined using a Seahorse XFe96 Analyzer (Agilent) according to the manufacturer's instructions. FACS-purified naive ($3 \times 10^5$ per well) or in vitro-activated ($2 \times 10^5$ per well) WT and KO p14 cells were seeded onto Cell-Tak (Corning, no. 354240) pre-coated Seahorse 96-well plates (Agilent, no. 102416-100) by centrifugation. The following compounds were injected during the experiment at the indicated time points: oligomycin (2 µM, Sigma Aldrich, O4876), FCCP (Carbonyl cyanide-4-(trifluoromethoxy) phenylhydrazone, 1.5 µM, Abcam, ab120081), rotenone (1 µM, Sigma Aldrich, R8875), antimycin A (1 µM, Sigma Aldrich, A8674), glucose (10 mM, Sigma Aldrich, G7021), and 2-deoxy-glucose (100 mM, Sigma Aldrich, D3179). OCR and ECAR values represent the averages of 7–10 replicates from two independent experiments, and are depicted as mean +/− SEM. Parameters of glycolysis and OXPHOS were calculated according to the manufacturer's instructions.

**Western blotting analysis**. Total cellular extracts from organ lysates were prepared in lysis buffer supplemented with proteinase inhibitor (Roche) and phosphatase inhibitor (Thermo Scientific). Protein samples were denatured at 95 °C for 5 min and loaded on Mini-proteanTGX precast gels (BioRad) for SDS-PAGE. The gel was blotted onto the polyvinylidene difluoride membrane (BioRad). After incubation with the primary polyclonal rabbit-α-mouse TNIK antibody (Thermo Scientific) at 4 °C overnight, horseradish peroxidase (HRP)-conjugated goat-anti-rabbit IgG (Thermo Scinentific) was used for primary antibody labeling. Alternatively, monoclonal HRP-conjugated anti-mouse β-actin antibody was used (SIGMA). Specifically labeled protein bands were detected with Clarity Western ECL Blotting Substrate using enhanced chemiluminescence (BioRad).

**Lentivirus-based reporter assay**. The Tcf/Lef reporter assay was performed as described by Reya et al.[75]. Briefly, human HIV-specific CD8$^+$ T cells were cultured in a 96-well U-bottom plate in CTL media without antibiotics and transduced with TCF/LEF lentiviral particles expressing firefly-luciferase or negative control lentiviral particles ([25 MOI], Cignal Lenti TCF/LEF reporter [luc] kit; SABioscience) for 24 h at 37 °C in the presence of 2 U ml$^{-1}$ human recombinant IL-2 and 8 µg ml$^{-1}$ SureEntry transduction reagent (SABioscience). Cells were then washed and cultured in CTL media for 4 days. Luciferase activity was quantified by an Infinite 200 microplate reader (Tecan) upon addition of Luficerin reagent (Steady-Glo Luciferase Assay System;Promega) following the manufacturers instructions. CD27$^+$/CD27$^-$ luminescence ratio was calculated and normalized to the negative control condition.

**RT-qPCR**. RNA was extracted from FACS-purified p14 T cells using NucleoSpin RNA kit (Macherey-Nagel, USA), and cDNA was synthesized using High Capacity cDNA Reverse Transcription Kit (Applied Biosystems, USA) for quantitative real-time (qRT) PCR. The real-time primers (Supplementary Table 1) were designed using Primerquest Software (Integrated DNA Technologies). For qRT-PCR analysis, the synthesized cDNAs was amplified with specific gene primers using FasStart Universal SYBR Green 2× Master Mix (Roche). Raw values were normalized to a reference gene (*GAPDH*). Real-time PCR reactions were performed in two independent biological replicates, and including RNAse-free H$_2$O controls using ABI Prism 7900 Sequence Detection System (Applied Biosystems). The fold difference for each sample was calculated by the comparative Ct method.

**Next-generation sequencing**. The total RNA was extracted from WT and KO p14 T cells of naive, D6, and D80 immunized mice using the RNeasy Micro Kit (QIAGEN AG, Switzerland) according to the manufacturer's instructions. The total RNA was quality-checked on the Bioanalyzer instrument (Agilent Technologies, USA) using the RNA 6000 Pico Chip (Agilent, USA) and quantified by Fluorometry using the QuantiFluor RNA System (Promega, USA).

Libraries were generated from 10 ng of the total RNA using the SmartSeq2 Kit (Takara Bio, Japan). Libraries were quality-checked on the Fragment Analyzer (Advanced Analytical, Ames, IA, USA) using the High Sensitivity NGS Fragment Analysis Kit (DNF-474, Advanced Analytical). Samples were pooled to equal molarity. Each pool was quantified by PicoGreen Fluorometric measurement in order to adjust to 12pM and used for clustering on the HiSeq 2500 instrument (Illumina, San Diego, CA, USA). Samples were sequenced single-reads 50 bases using the HiSeq SBS Kit v4 (Illumina, USA) and primary data analysis was performed with the Illumina RTA v1.18.66.3.

**RNA-seq analysis to access differentially expressed genes**. The RNA-seq data were analyzed using the ArrayStar software v.13 (DNASTAR, USA). The level of gene expression was assessed after Reads Per Kilobase Million (RPKM) normalization and log2 transformation. The dataset was analyzed by two-way ANOVA. A volcano plot was generated to illustrate differentially expressed genes of WT and KO p14 T cells of D6 and D80 immunized mice by plotting *P*-value versus fold change. Genes with significant difference in their expression at $P < 0.05$ and fold differences ≥1.5 were selected. The heatmaps were generated according to the standard normal distribution of the values, and data were clustered using the standard Euclidean's method based on the average linkage. RNA-seq based expression profile of TNIK in selected immune cell subsets was assessed using Gene Skyline platform.

**Gene set enrichment, gene ontology, and pathway analysis**. Gene set enrichment analysis (GSEA) was performed using the GSEA software v3.0 (Broad-institute, USA). Gene networks and canonical pathways representing differentially expressed genes were identified using the Ariadne Genomics Pathway Studio software and mammalian database (Elsevier). The gene ontology (GO) enrichment takes a list of significantly expressed genes at different treatment conditions and groups them into functional hierarchies. The enrichment scores were calculated using chi-square test comparing the proportion of the gene list to the proportion of the background in the group. A value of 3 or higher corresponded to a significant over-expression ($P < 0.05$). Depicted: normalized enrichment score (NES) for the gene set. Peak at the beginning/end suggests significant enrichment in KO or WT, respectively. Value of each gene's correlation with the phenotype, KO.

In silico pathway analysis was performed using the Ariadne Genomics Pathway Studio software (Elsevier). This analysis predicts potential biological processes, pathways and molecules affected by differentially expressed genes. The functional analysis identified direct interactions between differentially expressed genes in order to facilitate an understanding beyond their regulatory networks.

**Statistical analysis**. Statistical analysis was performed using GraphPad Prism software v7.0 (GraphPad, USA). The data were analyzed using one-way or two-way ANOVA or Student's *t* test (one-tailed, two-tailed). Significant differences in Kaplan–Meier survival curves were determined using the log-rank test (two-tailed). Data are represented as means ± standard error of the mean (SEM) as indicated in the legend. $P < 0.05$ was considered statistically significant. $*P < 0.05$, $**P < 0.01$, $***P < 0.001$, and $****P < 0.0001$. Ns denotes nonsignificant.

**Software**. PRISM software was used for the statistical analysis and data visualization (http://www.graphpad.com). Flow cytometric data was analyzed using FloJo 9 and 10 (http://www.flowjo.com). ImageStream X analysis was performed using Amnis IDEAS software (http://www.emdmillipore.com). For bioinformatics analysis of the RNA-seq data ArrayStar Software v13 (http://www.dnastar.com), Ariadne Genomic Pathway Studio software (http://www.ariadnegenomics.com) and Gene set enrichment analysis software v3.0 (http://software.broadinstitute.org/gsea) were utilized.

**Reporting summary**. Further information on research design is available in the Nature Research Reporting Summary linked to this article.

## Data availability

The NGS data are available in the NCBI GEO database under the accession code: GSE127734. Gene Skyline platform of Immunological Genome Project (http://rstats.immgen.org/Skyline/skyline.html) was used for RNA-seq analysis. The source data are provided as a Source Data file. All other data and reagents will be made available by the corresponding authors upon reasonable requests.

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

## Acknowledgements

The research leading to these results received founding form the Swiss National Science Foundation (310030_17017) and in part by funding from the Immunotherapy Integrated Research Center at the Fred Hutchinson Cancer Research Center. The authors thank Anne-Laure Huguenin, Daniela Korner, and Tanja Chiorazzo for their technical support, and Daniel D. Pinschewer for the scientific inputs and kindly providing us rVV-G2. We are grateful having received human HIV-1 Gag-specific T-cell clones generated at the Fred Hutchinson Cancer Research Center, Seattle, USA.

## Author contributions

Conceptualization: A.F.O. and C.A.J.-R. Methodology: A.F.O., C.A.J.-R., C.R., C.M.S. and S.F. Formal analysis: C.A.J.-R., S.H. and M.H. Investigation: C.A.J.-R., S.H., M.H., O.F., U.L. and M.F.A. Mouse generation and breedings: C.A.J.-R., C.R., C.M.S., E.D.B. and M.A.A. RNA-seq: R.R. and C.A.J.-R. Antibody engineering: C.E.C. Writing: C.A.J.-R. and A.F.O. All the listed authors have agreed all of the contents and approved the submitted version.

## Competing interests

The authors declare no competing interests.
