## [Peer Review File · Nature Communications]

Reviewers' comments:

Reviewer #1 (CD8 memory, transcriptomics)(Remarks to the Author):

In the manuscript "TNIK signaling imprints CD8 T cell memory formation early after priming" the authors investigate the role of tumor necrosis factor superfamily receptor signaling and activation of the TRAF2-/NCK-interacting kinase (TNIK) during memory T cell differentiation. The authors use the LCMV murine model system of acute viral infection to track antigen-specific CD8 T cell effector and memory differentiation in mice that conditionally delete TNIK. Deletion was driven by the tamoxifen inducible UBC cre. The authors demonstrate that absence of TNIK during the priming stage of the immune response resulted in normal effector responses, but a decline in the generation of memory T cells. This observation was first noted in endogenous cells and confirmed using TCR transgenic P14 cells, documenting the T-cell intrinsic nature of TNIK-deletion on the impaired memory differentiation. The authors also noted that development of the memory-precursor effector cell subset was coupled to increased TNIK expression. To further explore the biological pathways impacted by TNIK deletion in CD8 T cells the authors performed gene expression profiling using WT and KO T cells at effector and memory stages of the immune response. Based on insights from the gene expression analyses the authors then proceed to explore the role TNIK signaling in establishing asymmetric cell division of the T cells during the initial priming. The authors explore the ability of the WT and KO T cells to undergo symmetric and asymmetric cell division after in vitro co-culture with bone marrow derived DCs (assuming peptide loaded, but these details are absent). Lastly, the authors report that TNIK signaling in human HIV-specific CD8 T cells is coupled to CD27 expression and the beta-catenin pathway. However, the human data falls short of connecting the results to the initial priming stage of the immune response, which appears to be a key aspect of the story. Overall, the biology described in this study is interesting, and the data are clearly presented. Thus, it is likely that the readership of Nature Communications will appreciate this work. However, I have listed below a few questions / comments, that should be addressed, in order to strengthen the claims of the manuscript.

Specific comments/questions:

One of the major claims in this manuscript is that the impact of TNIK on memory differentiation is observed at the priming stage. A key experiment addressing the temporal relationship of TNIK and memory differentiation are stages of the immune response when deletion was performed. The authors state that deletion of TNIK at later stages of the immune response did not impact on memory differentiation, however this reviewer was unable to find evidence that the authors confirmed that deletion at later stages of the immune response is efficient. The authors need to purify the memory cells after tamoxifen induced deletion is performed at later time points and show that the locus is deleted.

Related to the timing deletion experiments discussed above, does late TNIK deletion (after the priming stage) impact on secondary effector and memory differentiation? Given that the manuscript is focused on the impact that the initial priming stage has on the commitment to effector or memory fates (symmetric vs asymmetric division) this reviewer is left wondering if TNIK impacts on the fate potential of memory cells when they re-encounter antigen.

The authors make some very strong claims regarding memory differentiation as it relates to symmetric versus asymmetric cell division based on their in vitro experiments. In general these in vitro studies fail to capture all of the interactions that CD8 T cells experience during the early in vivo priming stages of an immune response. The authors should consider significantly softening their claims with regard to the relationship between asymmetric cell division and bona fide memory differentiation. Specifically, this reviewer could find no in vivo asymmetric cell division experiments performed in this manuscript that can support the following claim on page 11 line #290 "These results suggest that CD27/TNIK signaling induces Wnt pathway activation, SD and

favors memory T cell differentiation “.

The HIV-specific CD8 T cell analyses are underdeveloped. This figure feels like an afterthought and the methods section does not adequately describe the source of the T cells, the authors simply cite their prior work (note the citation is also not formatted the same as the other citations – last minute addition to the story?).

The supplemental data and methods section describing the serial transfer and re-challenge experiments list the immunization dose of LCMV being 10,000 pfu of WE-LCMV. Is this still considered an acute infection? The authors need to show viral titers of the mice and demonstrate that the viral load peaked at the effector stage of the immune response and was then absent at the memory stage.

Minor Comments

Typo on line 42 of supplemental figure 3. “LCVM” should probably be removed from the sentence.

Reviewer #2 (CD8 memory, viral infection)(Remarks to the Author):

In this manuscript by Jaeger-Ruckstuhl et al., authors have investigated the role of TRAF2- and NIK-interacting kinase (TIK) in regulating the effector and memory CD8 T cell responses to LCMV. Studies using a global inducible KO showed that TIK deficiency did not affect the clonal expansion, effector function or differentiation of effector T cells during an acute LCMV infection. While the frequencies of GP33-specific CD8 T cells in WT and KO mice remained comparable at all timepoints, the absolute numbers of GP33-specific memory CD8 T cells were significantly reduced at day 80 after infection. And, a slightly greater percentage of KO memory T cells displayed a TEM phenotype. Authors also show that KO memory CD8 T cells exhibit impairment in secondary expansion in adoptively transferred hosts. Defects in the number of memory CD8 T cells and their recall responses were not seen if TIK was ablated 20 days after LCMV infection. This suggested that TIK regulates memory numbers and their function during 0-20 days after infection.

To examine cell-intrinsic effects, authors generate TIK-deficient P14 CD8 T cells. In a competitive adoptive transfer experiment (1:1 ratio of WT and KO P14 CD8 T cells), KO CD8 T cells seem to show greater proliferation early (Day 3) and express higher levels of T-bet (not EOMES). In a subsequent non-competitive experiment, authors show that TIK deficiency leads to fewer number of memory T cells and these cells display poor recall responses in adoptively transferred hosts. TIK-deficient secondary memory P14 CD8 T cells also show defective recall responses. Overall, these studies show that TIK promotes the development of memory T cells and is required for their regenerative capacity.

RNA sequencing showed altered expression of several genes in effector and memory TIK KO CD8 T cells. Specifically, genes regulating metabolism, stemness, cell cycle, cell death and immune signaling were altered in day 6 effector cells. Based on RNA seq data, conclusion is made that TIK deficiency leads to increased proliferation, differentiation towards effector cells, apoptosis and metabolic programming towards increased glycolysis. In TIK KO memory T cells, genes involved in catabolic and anabolic processes, mitochondrial biogenesis, oxidative stress and protein synthesis were altered.

Lastly, using an in vitro stimulation assay, authors examine whether TIK regulated symmetric and asymmetric cell division in CD8 T cells. They find that expression of Wnt signaling genes such as TCF7, Lef1 and Myc and stemness-related gene such as Msi2 were lower in KO P14 CD8 T cells.

They also found that TNIK deficiency enhanced asymmetric cell division at the expense of symmetric cell division. Lastly, data is provided to demonstrate that CD27 signaling-induced Wnt activation and symmetric cell division require TNIK.

Overall Summary

This manuscript is well written and addresses the role of TNIK signaling in regulating the development and function of effector and memory CD8 T cells. Overall the experiments are done well and show convincingly that TNIK deficiency leads to: (1) the development of fewer memory CD8 T cells; (2) reduced regenerative potential of memory CD8 T cells. Based on RNA seq data and in vitro experiments, authors propose that TNIK signaling promotes memory formation by augmenting symmetric cell division and facilitating CD27-induced Wnt signaling in CD8 T cells.

Overall Critique

The finding that TNIK controls recall responses of memory CD8 T cells is very interesting. This is the major finding of the study, but authors make no attempt to determine the mechanistic basis of this defect. RNA seq data provides new information, but no follow-up experiments are performed. Authors show that Wnt signaling including the expression of TCF7 and Lef1 is reduced in TNIK KO T cells. Despite the well-established role for TCF-1 in stemness and memory function, authors fail to investigate if TCF-1 complementation rectifies memory defects in TNIK KO CD8 T cells. Likewise, RNA seq data suggest metabolic alterations in TNIK KO CD8 T cells, but no attempt is made to investigate whether loss of metabolic fitness and loss of spare respiratory capacity could explain defects in TNIK KO memory CD8 T cells. No convincing data is provided that altered symmetric division contributes to fewer TNIK memory CD8 T cells. In summary, lack of mechanistic experiments to explain major findings undermines the significance of the manuscript.

Specific Comments/Concerns

Many conclusions in the manuscript are not supported by data. For example, except for increased T-bet expression, no data is provided to show that TNIK deficiency actually promotes effector differentiation. Where is the data on differentiation of SLECs and MPECs and expression of canonical transcription factors in the competitive P14 model? Why was memory development not studied using the competitive P14 model? Authors also conclude that TNIK KO CD8 T cells underwent more apoptosis during expansion without providing convincing data (Fig. 3c). What percentages of WT and KO P14 CD8 T were Annexin V positive? Did they confirm by quantifying caspase 3 or Bim/Bcl-2 levels?

TNIK KO memory CD8 T cells show delayed proliferation and markedly reduced accumulation of secondary/tertiary effectors. Is this due to metabolic dysregulation or enhanced apoptosis or enhanced function of cell cycle inhibitors? In studies of primary response (Figure 3) and recall responses, authors need to carefully quantify both proliferation (Ki67) and apoptosis (Annexin V) rates.

Important findings from RNAseq data are not confirmed by RT-PCR or protein analysis (Example: BATF in memory CD8 T cells). Data from RNAseq analysis provided new insights on metabolism and proliferation, but authors did not follow-up on this data. Studies of metabolism such as glycolysis and oxidative phosphorylation are warranted.

Studies of asymmetric/symmetric cell division incorporating molecules such as T-bet, IRF-4 and TCF-1 would better support the conclusions.

While in vitro studies show reduced expression of Tcf7, Lef1 and Myc in TNIK KO CD8 T cells. Surprisingly, the expression of these molecules wasn't confirmed in P14 CD8 T cells at different days PI.

Point-by-point reply

Reviewer #1 (CD8 memory, transcriptomics)(Remarks to the Author):

In the manuscript “TNIK signaling imprints CD8 T cell memory formation early after priming” the authors investigate the role of tumor necrosis factor superfamily receptor signaling and activation of the TRAF2-/NCK-interacting kinase (TNIK) during memory T cell differentiation. The authors use the LCMV murine model system of acute viral infection to track antigen-specific CD8 T cell effector and memory differentiation in mice that conditionally delete TNIK. Deletion was driven by the tamoxifen inducible UBC cre. The authors demonstrate that absence of TNIK during the priming stage of the immune response resulted in normal effector responses, but a decline in the generation of memory T cells. This observation was first noted in endogenous cells and confirmed using TCR transgenic P14 cells, documenting the T-cell intrinsic nature of TNIK-deletion on the impaired memory differentiation. The authors also noted that development of the memory-precursor effector cell subset was coupled to increased TNIK expression. To further explore the biological pathways impacted by TNIK deletion in CD8 T cells the authors performed gene expression profiling using WT and KO T cells at effector and memory stages of the immune response. Based on insights from the gene expression analyses the authors then proceed to explore the role TNIK signaling in establishing asymmetric cell division of the T cells during the initial priming. The authors explore the ability of the WT and KO T cells to undergo symmetric and asymmetric cell division after in vitro co-culture with bone marrow derived DCs (assuming peptide loaded, but these details are absent). Lastly, the authors report that TNIK signaling in human HIV-specific CD8 T cells is coupled to CD27 expression and the beta-catenin pathway. However, the human data falls short of connecting the results to the initial priming stage of the immune response, which appears to be a key aspect of the story. Overall, the biology described in this study is interesting, and the data are clearly presented. Thus, it is likely that the readership of Nature Communications will appreciate this work. However, I have listed below a few questions / comments, that should be addressed, in order to strengthen the claims of the manuscript.

Specific comments/questions:

RQ (1):

One of the major claims in this manuscript is that the impact of TNIK on memory differentiation is observed at the priming stage. A key experiment addressing the temporal relationship of TNIK and memory differentiation are stages of the immune response when deletion was performed. The authors state that deletion of TNIK at later stages of the immune response did not impact on memory differentiation, however this reviewer was unable to find evidence that the authors confirmed that deletion at later stages of the immune response is efficient. The authors need to purify the memory cells after tamoxifen induced deletion is performed at later time points and show that the locus is deleted.

Response RQ (1):

The authors have assessed *Tnik* gene expression of total peripheral blood cells 10 days post last tamoxifen administration. Setting the mark for efficient TNIK deletion at 90% or higher (relative to lowest WT value), 7 out of 10 tamoxifen treated *Tnik*^{F/F} mice fulfilled the inclusion criteria, whereas 3 out of 10 mice (N11, P3 and P30) were excluded from the experiment (Supplementary Fig. 3b, left bar graph). In order to confirm constitutive *Tnik* deletion, we assessed gene expression levels of FACS-purified gp33⁺ memory CD8⁺ T cells isolated from spleens 80 days post immunisation. Data confirm that tamoxifen induced *Tnik* deletion was durable (Supplementary Fig. 3b, right bar graph).

RQ (2):

Related to the timing deletion experiments discussed above, does late TNIK deletion (after the priming stage) impact on secondary effector and memory differentiation? Given that the manuscript is focused on the impact that the initial priming stage has on the commitment to effector or memory fates (symmetric vs asymmetric division) this reviewer is left wondering if TNIK impacts on the fate potential of memory cells when they re-encounter antigen.

Response RQ (2):

To answer this question, we immunized *Tnik*^{WT} and *Tnik*^{F/F} mice that underwent tamoxifen-induced deletion before (PRE, Δ/Δ) immunization or after priming (POST, Δ/Δ_{20}) with 200 pfu LCMV-WE (1st). 30 days post primary immunization, mice were re-challenged with 10⁶ pfu recombinant vaccinia virus expressing the glycoprotein of LCMV (rVV-G2) i.p.. Tcf1 is an essential transcription factor for self-renewal of T cells (Kratchmarov R. et al., Blood Adv. 2018 Jul 24; 2(14): 1685–1690). We have assessed Tcf1 expression to evaluate memory fate potential during primary and secondary effector and memory differentiation. We found that Tcf1 expression was not significantly changed in WT vs Δ/Δ gp33-specific CD8 T_{EFF} cells on day 8 p.i., although there was a trend for a lower Tcf1 expression in Δ/Δ gp33-specific CD8 T_{EFF} cells (Fig. 1g). However, Tcf1 expression was strongly reduced in Δ/Δ T_{CM} and T_{EM} memory cells day 30 p.i. (Fig. 2b). Re-challenge with rVV-G2 revealed that Δ/Δ cells have a reduced frequency of Klrg1⁺Tcf1⁺ secondary effector cells (day 4 p.i, Fig 2g). The frequency of CD127⁺Tcf1⁺gp33⁺CD8⁺ T cells was higher in WT vs Δ/Δ 38 days post re-challenge (Fig. 2h) and gp33⁺CD8⁺ T cells expressed lower levels of Tcf1 per cell (Fig. 2i,j) as shown in blood and spleen. This let us suggest that Tcf1 expression was better maintained in the gp33⁺CD8⁺ WT vs Δ/Δ

compartment during primary (1st) and secondary (2nd) immune response. In contrast, *Tnik* deletion during primary contraction phase (Δ/Δ_{20}) did not affect primary memory and secondary effector differentiation, however secondary memory development became affected. The fraction of CD127⁺Tcf1⁺ memory cells was reduced in Δ/Δ_{20} mice and gp33⁺CD8⁺ memory T cells expressed less Tcf1 in Δ/Δ_{20} mice (Supplementary Fig. 3d, 3i-l).

We conclude that TNIK deletion after priming (Δ/Δ_{20}) does not impact primary memory (1st) formation and expansion after antigen re-challenge. However, secondary memory formation after antigen re-exposure is impaired. This indicates that TNIK is required for memory formation during T cell stimulation and expansion.

RQ (3):

The authors make some very strong claims regarding memory differentiation as it relates to symmetric versus asymmetric cell division based on their in vitro experiments. In general these in vitro studies fail to capture all of the interactions that CD8 T cells experience during the early in vivo priming stages of an immune response. The authors should consider significantly softening their claims with regard to the relationship between asymmetric cell division and bona fide memory differentiation.

Response RQ (3):

As requested, we only mention the relationship between SD and AD and memory formation as a hypothesis in the revised manuscript but not as a conclusion derived from our experiments (Page 12, line 6-9 of the revised manuscript). In addition, we added data that confirm the relevance of AD and SD during T cell activation in vivo (Fig 6f).

RQ (4):

Specifically, this reviewer could find no in vivo asymmetric cell division experiments performed in this manuscript that can support the following claim on page 11 line #290 “These results suggest that CD27/TNIK signaling induces Wnt pathway activation, SD and favors memory T cell differentiation “.

Response RQ (4):

In order to gain more evidence that signaling via TNIK promotes symmetric segregation of the fate determinant Numb *in vivo*, we FACS-purified adoptively transferred WT and KO p14 T cells from spleen 4 days after immunization with LCMV. This experiment confirmed our *in vitro* experiments and documented an increase in AD in the absence of *Tnik in vivo* (Fig. 6f).

RQ (5):

The HIV-specific CD8 T cell analyses are underdeveloped. This figure feels like an afterthought and the methods section does not adequately describe the source of the T cells, the authors simply cite their prior work (note the citation is also not formatted the same as the other citations – last minute addition to the story?).

Response RQ (5):

The authors have expanded HIV1- gag-specific CD8⁺ T cell analysis with live imaging of activated (α CD3/ α CD27 vs α CD3/ α CD28) naïve human CD8⁺ T cells and analyzed segregation of mitochondria in dividing daughter cells. The data

show that CD27 co-stimulation drives more symmetric mitochondrial segregation and maintenance of Tcf1 expression. Daughter cells with low mitochondrial content may undergo faster differentiation due to lack of metabolic fitness. We have included the data in the manuscript (Fig. 7e-g). We think it is important to report on some confirmatory experiments with human T cells. The data section describes TNIK/ β -catenin/WNT signaling in human HIV-1 gag-specific CD8⁺ T cells during re-stimulation, and further confirm that also lack of CD27 co-stimulation during re-challenge impacts maintenance of T cells with high Wnt activity (see Supplementary Fig. 3j-l). In addition, we have modified the methods section describing the source and generation of HIV-1 gag-specific CD8⁺ T cells, have added an additional citation and corrected the citation formatting error.

RQ (6):

The supplemental data and methods section describing the serial transfer and re-challenge experiments list the immunization dose of LCMV being 10,000 pfu of WE-LCMV. Is this still considered an acute infection? The authors need to show viral titers of the mice and demonstrate that the viral load peaked at the effector stage of the immune response and was then absent at the memory stage.

Response RQ (6):

We added the requested data in the revised manuscript. Mice adoptively transferred with 10⁵ p14 T cells and primed with 10⁴ pfu LCMV WE virus, show high virus titers at day 6 p.i. and completely eliminate the virus by D60 p.i. independent of the expression of Tnik (Supplementary Fig. 4f-g).

Minor Comments:

Typo on line 42 of supplemental figure 3. "LCVM" should probably be removed from the sentence.

We have corrected the spelling error in supplemental Fig.3 of the revised manuscript.

Reviewer #2 (CD8 memory, viral infection)(Remarks to the Author):

In this manuscript by Jaeger-Ruckstuhl et al., authors have investigated the role of TRAF2- and NICK-interacting kinase (TNIK) in regulating the effector and memory CD8 T cell responses to LCMV. Studies using a global inducible KO showed that TNIK deficiency did not affect the clonal expansion, effector function or differentiation of effector T cells during an acute LCMV infection. While the frequencies of GP33-specific CD8 T cells in WT and KO mice remained comparable at all timepoints, the absolute numbers of GP33-specific memory CD8 T cells were significantly reduced at day 80 after infection. And, a slightly greater percentage of KO memory T cells displayed a TEM phenotype. Authors also show that KO memory CD8 T cells exhibit impairment in secondary expansion in adoptively transferred hosts. Defects in the number of memory CD8 T cells and their recall responses were not seen if TNIK was ablated 20 days after LCMV infection. This suggested that TNIK regulates memory numbers and their function during 0-20 days after infection.

To examine cell-intrinsic effects, authors generate TNIK-deficient P14 CD8 T cells. In a competitive adoptive transfer experiment (1:1 ratio of WT and KO P14 CD8 T cells), KO CD8 T cells seem to show greater proliferation early (Day 3) and express higher levels of T-bet (not EOMES). In a subsequent non-competitive experiment, authors show that TNIK deficiency leads to fewer number of memory T cells and these cells display poor recall responses in adoptively transferred hosts. TNIK-deficient secondary memory P14 CD8 T cells also show defective recall responses. Overall, these studies show that TNIK promotes the development of memory T cells and is required for their regenerative capacity.

RNA sequencing showed altered expression of several genes in effector and memory TNIK KO CD8 T cells. Specifically, genes regulating metabolism, stemness, cell cycle, cell death and immune signaling were altered in day 6 effector cells. Based on RNA seq data, conclusion is made that TNIK deficiency leads to increased proliferation, differentiation towards effector cells, apoptosis and metabolic programming towards increased glycolysis. In TNIK KO memory T cells, genes involved in catabolic and anabolic processes, mitochondrial biogenesis, oxidative stress and protein synthesis were altered.

Lastly, using an in vitro stimulation assay, authors examine whether TNIK regulated symmetric and asymmetric cell division in CD8 T cells. They find that expression of Wnt signaling genes such as TCF7, Lef1 and Myc and stemness-related gene such as Msi2 were lower in KO P14 CD8 T cells. They also found that TNIK deficiency enhanced asymmetric cell division at the expense of symmetric cell division. Lastly, data is provided to demonstrate that CD27 signaling-induced Wnt activation and symmetric cell division require TNIK.

Overall Summary

This manuscript is well written and addresses the role of TNIK signaling in regulating the development and function of effector and memory CD8 T cells. Overall the experiments are done well and show convincingly that TNIK deficiency leads to: (1) the development of fewer memory CD8 T cells; (2) reduced regenerative potential of memory CD8 T cells. Based on RNA seq data and in vitro experiments, authors propose that TNIK signaling promotes memory formation by augmenting symmetric cell division and facilitating CD27-induced Wnt signaling in CD8 T cells.

Overall Critique

The finding that TNIK controls recall responses of memory CD8 T cells is very interesting. This is the major finding of the study, but authors make no attempt to determine the mechanistic basis of this defect. RNA seq data provides new information, but no follow-up experiments are performed. Authors show that Wnt signaling including the expression of TCF7 and Lef1 is reduced in TNIK KO T cells. Despite the well-established role for TCF-1 in stemness and memory function, authors fail to investigate if TCF-1 complementation rectifies memory defects in TNIK KO CD8 T cells. Likewise, RNA seq data suggest metabolic alterations in TNIK KO CD8 T cells, but no attempt is made to investigate whether loss of metabolic fitness and loss of spare respiratory capacity could explain defects in TNIK KO memory CD8 T cells. No convincing data is provided that altered symmetric division contributes to fewer TNIK memory CD8 T cells. In summary, lack of mechanistic experiments to explain major findings undermines the significance of the manuscript.

Specific Comments/Concerns

RQ (7):

Many conclusions in the manuscript are not supported by data. For example, except for increased T-bet expression, no data is provided to show that TNIK deficiency actually promotes effector differentiation. Where is the data on differentiation of SLECs and MPECs and expression of canonical transcription factors in the competitive P14 model? Why was memory development not studied using the competitive P14 model? Authors also conclude that TNIK KO CD8 T cells underwent more apoptosis during expansion without providing convincing data (Fig. 3c). What percentages of WT and KO P14 CD8 T were Annexin V positive? Did they confirm by quantifying caspase 3 or Bim/Bcl-2 levels?

Response RQ (7):

As requested, we performed additional experiments assessing the differentiation of D10 AdCoTf p14 T cells. We could not detect significant changes in MPECs (Klrg1⁻CD127⁺) and SLECs (Klrg1⁺CD127⁻) frequencies between WT vs KO cells (Supplementary Fig. 4e). The canonical transcription factors T-bet and Eomes were already included in the previous version of the manuscript (see Fig. 3f).

As requested, we now included additional evidence of TNIK-dependent memory fate commitment by analyzing Tcf1 expression in gp33⁺ effector and memory T cells using the conditional depletion model. We now show that TNIK-deficiency during priming (Δ/Δ) did not affect primary effector differentiation but resulted in reduced maintenance of Tcf1⁺ (T_{CM} + T_{EM}) memory cells (Fig. 2b). After re-challenge with LCMV-GP expressing vaccinia virus (Vacc-G2), secondary Δ/Δ effectors showed reduced frequencies of Tcf1⁺Klrg1⁺ effector and Tcf1⁺CD127⁺ memory T cells (Fig. 2g-k). These data underline that TNIK signaling during priming is crucial for formation and maintenance of Tcf1⁺ memory cells.

Further, the authors were asked to provide frequencies of AnnexinV⁺ stained WT and KO p14 CD8 T cells in the competitive transfer model. This data is now included in the revised manuscript (Supplementary Fig. 4b). Tnik KO cells have a higher frequency of AnnexinV⁺ cells d3, d7 and d10 p.i.

We did not quantify caspase 3 or Bim/Bcl-2 on protein levels in the AdCoTf model. However, RNAseq data obtained from the non-competitive AdTf model

(Fig. 4e) provide a trend that pro-apoptotic genes (Bak1, Bcl10, Bik, Xiap, Casp9) are upregulated and anti-apoptotic genes (Bcl2, Bcl2l1) are downregulated in D6 KO versus WT P14 T cells (Supplementary Fig. 5i). However, although none of the analyzed genes, except *Bcl10*, was changed significantly, the GSEA analysis (HALLMARK_APOPTOSIS) indicated a significant change to a pro-apoptotic gene signature (Fig 4e). New data provided in RQ(10) support the notion that day 2 effector KO vs WT p14 T cells show a trend towards increased Casp3/Bim/Casp9 gene expression (Supplementary Fig. 7b). New data provided in RQ(8) support the notion that KO p14 T cells undergo more apoptosis during primary contraction, secondary expansion and secondary contraction than WT p14 T cells (Supplementary Fig. 4j).

RQ (8):

TNIK KO memory CD8 T cells show delayed proliferation and markedly reduced accumulation of secondary/tertiary effectors. Is this due to metabolic dysregulation or enhanced apoptosis or enhanced function of cell cycle inhibitors? In studies of primary response (Figure 3) and recall responses, authors need to carefully quantify both proliferation (Ki67) and apoptosis (Annexin V) rates.

Response RQ (8):

We have performed primary, secondary and tertiary immunization experiments to carefully quantify proliferation and apoptosis using Ki67⁺ and AnnexinV⁺ staining rates, respectively.

AnnexinV⁺ fractions were found to be higher in KO p14 T cells during primary contraction phase, secondary effector and contraction phase (Supplementary Fig. 4j). In contrast, Ki67 staining did not provide evidence that proliferation was altered between WT and KO p14 T cells during primary or secondary expansion. Our results indicate that at any of the timepoints analyzed between pre-peak expansion (day 5 p.i.) and early contraction (day 7 & 10 p.i.), a majority of virus-specific cells in the blood were Ki67⁺(G1,S,G2,M) (Supplementary Fig. 4k). The question regarding metabolic regulation of WT and KO p14 T cells was assessed and answers can be found under RQ(9).

RQ (9):

Important findings from RNAseq data are not confirmed by RT-PCR or protein analysis (Example: BATF in memory CD8 T cells). Data from RNAseq analysis provided new insights on metabolism and proliferation, but authors did not follow-up on this data. Studies of metabolism such as glycolysis and oxidative phosphorylation are warranted.

Response RQ (9):

As requested, we confirmed RNAseq data by assessing RTqPCR of selected genes, such as *Notch1*, *Eomes*, *Nfatc1*, *Map2k5*, *Lig12* for D6 WT and KO p14 T cells and *Batf*, *CD74*, *Nfkb2*, *Rora*, *Bcl11b* for D80 WT and KO p14 T cells (Supplementary Fig. 5b).

To extend the results on proliferation we have analyzed 3h BrdU incorporation and CFSE dilution of WT and KO p14 T cells *in vivo* (AdCoTf): BrdU incorporation rate day 3 p.i. as well as CFSE dilution day 4 p.i. show that KO p14 T cells proliferate slightly more compared to WT p14 T cells, however not to a significant extent (Supplementary Fig. 4c,d).

We also analyzed the metabolic profile of naïve (reflecting steady-state) and day 3 H8-DC *in vitro* activated WT and KO p14 CD8⁺ T cells using Seahorse technology. Evidence is provided, that naïve WT p14 T cells have a higher maximal respiratory capacity, spare respiratory capacity, ATP-linked respiration, glycolytic capacity and glycolytic reserve capacity than KO p14 T cells (Fig. 4g). Interestingly, primary activated KO p14 T cells have a higher spare respiratory capacity and short-term glycolytic potential, indicating that they are more effector-like differentiated (Fig. 4h and Supplementary Fig. 5m). Therefore, RNAseq data suggested an influence on cell cycling, increased apoptosis and differences in metabolic activity in the absence of Tnik during priming. Functionally, we could confirm a role of apoptosis and metabolism that may explain the reduced memory capacity of Tnik KO cells, while a reduction in proliferation seems less important. This is discussed in the revised manuscript on page14.

RQ (10):

Studies of asymmetric/symmetric cell division incorporating molecules such as T-bet, IRF-4 and TCF-1 would better support the conclusions.

While in vitro studies show reduced expression of Tcf7, Lef1 and Myc in TNIK KO CD8 T cells. Surprisingly, the expression of these molecules wasn't confirmed in P14 CD8 T cells at different days PI.

Response RQ (10):

Studies of asymmetric/symmetric cell division were expanded by documenting an increased AD *in vivo* in the absence of Tnik (Fig. 6f). In addition, AD/SD was studied by analyzing mitochondrial segregation in naïve *in vitro* activated CD8⁺ T cells. This data is now included in the revised manuscript (Fig. 7e,f). In order to confirm our *in vitro* data shown in Fig. 6a also *in vitro*, we AdTf 10⁶ p14 T cells into mice prior to LCMV infection [10⁴ pfu LCMV]. We were able to FACS-purify sufficient numbers of p14 T cells from spleens at 48h p.i. (but not earlier) and assessed gene expression of selected genes using RTqPCR. We provide evidence that *Lef1* gene expression is reduced in KO vs WT p14 T cells 48h p.i.. *Tcf7* and *Myc* gene expression was not differentially expressed between purified WT and KO cells. However, we also show that KO p14 T cells have a general trend towards apoptosis (Casp3, Bim, Casp9). We have included the data in the revised manuscript (Supplementary Fig. 7b).

REVIEWERS' COMMENTS:

Reviewer #1 (Remarks to the Author):

The authors have sufficiently addressed the concerns previously raised by this reviewer. This reviewer is particularly excited by the results from the secondary effector studies as well as the interpretation.

Reviewer #2 (Remarks to the Author):

Authors provide evidence that TNIK signaling is required for recall responses of memory CD8 T cells in secondary and tertiary responses. This is an important discovery and of importance to the field of T cell memory. Authors show that there is increased apoptosis of TNIK-deficient T cells during the expansion/early contraction, and how this is linked to the imprinting of memory T cell recall responses is unclear; this needs to be discussed and explained. Also, it is unclear how metabolic alterations in TNIK-deficient T cells leads to reduced accumulation of T cells. Authors need to integrate all the findings into a cohesive hypothesis in the discussion.

REVIEWERS' COMMENTS:

Reviewer #1 (Remarks to the Author):

The authors have sufficiently addressed the concerns previously raised by this reviewer. This reviewer is particularly excited by the results from the secondary effector studies as well as the interpretation.

Reviewer #2 (Remarks to the Author):

Authors provide evidence that TNK1 signaling is required for recall responses of memory CD8 T cells in secondary and tertiary responses. This is an important discovery and of importance to the field of T cell memory. Authors show that there is increased apoptosis of TNK1-deficient T cells during the expansion/early contraction, and how this is linked to the imprinting of memory T cell recall responses is unclear; this needs to be discussed and explained. Also, it is unclear how metabolic alterations in TNK1-deficient T cells leads to reduced accumulation of T cells. Authors need to integrate all the findings into a cohesive hypothesis in the discussion.

The discussion has been modified.

TNK1 favours the differentiation to terminally differentiated effector cells. These cells usually undergo apoptosis after exertion of effector function. Thus, the increased levels of apoptosis detected in TNK1-deficient T cells support our hypothesis of increased terminal differentiation. This is added now in the discussion section on page 15. Moreover, increased glycolysis is a hallmark of effector cell differentiation. This was already included in the manuscript on page 14 of the discussion. Mechanistically, we documented that glycolysis is regulated by TNK1 via the PI3K/Akt pathway. This is discussed on page 14 of the Discussion section.

THANK YOU!